# Lagrangian Proximal Gradient Descent for Training Architectures with Embedded Optimization Layers

## Abstract

We propose *Lagrangian Proximal Gradient Descent* (LPGD), a flexible framework for embedding parameterized optimization layers into machine learning architectures trained via gradient backpropagation. Such layers often have degenerate derivatives, e.g. in the case of discrete optimization problems. Our method efficiently computes meaningful replacements of these derivatives by re-running the forward solver oracle on a perturbed input, capturing various previously proposed methods as special cases. LPGD is derived as gradient descent on the envelope of a loss linearization. Such interpretation fosters deep links between traditional and contemporary optimization methods. We prove that LPGD updates converge to the true gradient as the smoothening parameter approaches zero. Finally, we show experimentally on synthetic data that LPGD converges faster than gradient descent, even when non-degenerate derivatives exist.

## 1 Introduction

Optimization at inference is inherent to many prediction tasks, including autonomous driving (Paden et al., 2016), modelling physical systems (Cranmer et al., 2020), or robotic control (Kumar et al., 2016). Therefore, embedding optimization algorithms as building blocks of machine learning models serves as a powerful inductive bias. A recent trend has been to embed convex optimization problems that can efficiently be solved to optimality (Amos & Kolter, 2017a; Agrawal et al., 2019a;b; Vlastelica et al., 2020; Sun et al., 2022; Sahoo et al., 2023).

Training such a *parameterized optimization model* is an instance of bi-level optimization (Gould et al., 2016), which is generally challenging. Whenever it is possible to propagate gradients through the optimization problem via an informative derivative of the solution mapping, the task is typically approached with standard stochastic gradient descent (GD) (Amos & Kolter, 2017a; Agrawal et al., 2019b). However, when the optimization problem has discrete solutions, the derivatives are typically not informative, as small perturbations of the input do not affect the optimal solution. Previous works have proposed several methods to overcome this challenge, ranging from differentiable relaxations (Wang et al., 2019; Wilder et al., 2019a; Mandi & Guns, 2020; Djolonga & Krause, 2017) and stochastic smoothing (Berthet et al., 2020; Dalle et al., 2022), over proxy losses (Paulus et al., 2021), to finite-difference based techniques (Vlastelica et al., 2020).

The main contribution of this work is the **unification of a variety of previous methods** (McAllester et al., 2010; Vlastelica et al., 2020; Domke, 2010; Sahoo et al., 2023; Elmachtoub & Grigas, 2022; Blondel et al., 2020) into a **general framework** called *Lagrangian Proximal Gradient Descent* (LPGD). It is motivated by traditional proximal optimization techniques (Moreau, 1962; Rockafellar, 1970; Nesterov, 1983; Figueiredo et al., 2007; Tseng, 2008; Beck & Teboulle, 2009; Combettes & Pesquet, 2011; Bauschke & Combettes, 2011; Nesterov, 2014; Parikh & Boyd, 2014), thereby fostering deep links between traditional and contemporary methods.

When the derivatives of the solution mapping are degenerate, LPGD allows learning the optimization parameters even when GD fails, generalizing Vlastelica et al. (2020) to non-linear objectives, saddle-point problems, and learnable constraint parameters. When non-degenerate derivatives exist, we show that they can be computed as the limit of the LPGD update, which provides a fast and simple alternative to previous methods based on differentiating the optimality conditions (Amos & Kolter, 2017a; Agrawal et al., 2019b; Wilder et al., 2019a; Mandi & Guns, 2020). Finally, we show experimentally on synthetic data that LPGD results in faster convergence than GD, even when non-degenerate derivatives are available.

## 2 RELATED WORK

Numerous implicit layers have been proposed in recent years, including neural ODEs (Chen et al., 2018; Dupont et al., 2019) and root-solving-based layers (Bai et al., 2019; 2020; Gu et al., 2020; Winston & Kolter, 2020; Fung et al., 2021; Ghaoui et al., 2021; Geng et al., 2021). In this work, we focus on optimization-based layers. A lot of research has been done on obtaining the gradient of such a layer, either by using the implicit function theorem to differentiate quadratic programs (Amos & Kolter, 2017a), conic programs (Agrawal et al., 2019b), ADMM (Sun et al., 2022), dynamic time warping (Xu et al., 2023), or by finite-differences (Domke, 2010; McAllester et al., 2010; Song et al., 2016; Lorberbom et al., 2019).

Another direction of related work has investigated optimization problems with degenerate derivatives of the solution mapping. The techniques developed for training these models range from continuous relaxations of SAT problems (Wang et al., 2019) and submodular optimization (Djolonga & Krause, 2017), over regularization of linear programs (Amos et al., 2019; Wilder et al., 2019a; Mandi & Guns, 2020; Paulus et al., 2020) to stochastic smoothing (Berthet et al., 2020; Dalle et al., 2022), learnable proxies (Wilder et al., 2019b) and generalized straight-through-estimators (Jang et al., 2017; Sahoo et al., 2023). Other works have built on geometric proxy losses (Paulus et al., 2021) and, again, finite-differences (Vlastelica et al., 2020; Niepert et al., 2021; Minervini et al., 2023).

Finally, a special case of an optimization layer is to embed an optimization algorithm as the final component of the prediction pipeline. This encompasses energy-based models (LeCun & Huang, 2005; Blondel et al., 2022), structured prediction (McAllester et al., 2010; Blondel, 2019; Blondel et al., 2020), smart predict-then-optimize (Ferber et al., 2020; Elmachtoub & Grigas, 2022) and symbolic methods such as SMT solvers (Fredrikson et al., 2023). We present additional details of the closest related methods in Appendix B.

## 3 PROBLEM SETUP

We consider a parameterized embedded constrained optimization problem of the form

$$\mathcal{L}^*(w) := \min_{x \in \mathcal{X}} \max_{y \in \mathcal{Y}} \mathcal{L}(x, y, w) \tag{1}$$

where $w \in \mathbb{R}^k$ are the parameters, $\mathcal{X} \subseteq \mathbb{R}^n$ and $\mathcal{Y} \subseteq \mathbb{R}^m$ are the primal and dual feasible set, and $\mathcal{L}$ is a continuously differentiable *Lagrangian*. The corresponding optimal solution is

$$z^*(w) = (x^*(w), y^*(w)) := \arg\min_{x \in \mathcal{X}} \max_{y \in \mathcal{Y}} \mathcal{L}(x, y, w). \tag{2}$$

For instance, this setup covers conic programs and quadratic programs, see Appendix C for details. Note, that the solution of (2) is in general set-valued. We assume that the solution set is non-empty and has a selection $z^*(w)$ continuous at $w$.[1] Throughout the paper, we assume access to an oracle that efficiently solves (2) to high accuracy. In our experiments, (2) is a conic program which we solve using the SCS solver (O'Donoghue et al., 2016).[2]

Our aim is to embed optimization problem (2) into a larger prediction pipeline. Given an input $\mu \in \mathbb{R}^p$ (e.g. an image), the parameters of the embedded optimization problem $w$ are predicted by a parameterized backbone model $W_\theta \colon \mathbb{R}^p \to \mathbb{R}^k$ (e.g. a neural network with weights $\theta \in \mathbb{R}^r$) as $w = W_\theta(\mu)$. The embedded optimization problem (2) is then solved on the predicted parameters $w$ returning the predicted solution $x^*(w)$, and its quality is measured by a loss function $\ell \colon \mathbb{R}^n \to \mathbb{R}$. The backbone and the loss function are assumed to be continuously differentiable.

Our goal is to train the prediction pipeline by minimizing the loss on a dataset of inputs $\{\mu_i\}_{i=1}^N$

$$\min_{\theta \in \mathbb{R}^r} \sum_{i=1}^N \ell\big(x^*(W_\theta(\mu_i))\big) \tag{3}$$

using stochastic gradient descent or variations thereof (Kingma & Ba, 2015). However, the solution mapping does not need to be differentiable, and even when it is, the derivatives are often degenerate

---

[1]For example, these assumptions can be ensured by assuming compactness of $\mathcal{X}$ and $\mathcal{Y}$ and the existence of a unique solution, given the continuity of $\mathcal{L}$.

[2]We also use CVXPY (Diamond & Boyd, 2016; Agrawal et al., 2019a) which allows to automatically reduce parameterized convex optimization problems to parameterized conic programs in a differentiable way.

(e.g. they can be zero almost everywhere).[3] Therefore, we aim to derive informative replacements for the gradient $\nabla_w \ell(x^*(w))$, which can then be further backpropagated to the weights $\theta$ by standard automatic differentiation libraries (Abadi et al., 2015; Bradbury et al., 2018; Paszke et al., 2019). Note that the loss could also be composed of further learnable components that might be trained simultaneously. A list of all relevant symbols used in the main text is provided in Appendix H.

## 4 BACKGOUND: PROXIMAL GRADIENT DESCENT & MOREAU ENVELOPE

The *Moreau envelope* (Moreau, 1962) $\text{env}_{\tau f}\colon \mathbb{R}^n \to \mathbb{R}$ of a proper, lower semi-continuous, possibly non-smooth function $f\colon \mathbb{R}^n \to \mathbb{R}$ is defined for $\tau > 0$ as

$$\text{env}_{\tau f}(\widehat{x}) := \inf_x f(x) + \tfrac{1}{2}\|x - \widehat{x}\|_2^2. \tag{4}$$

The envelope $\text{env}_{\tau f}$ is a smoothed lower bound approximation of $f$ (Rockafellar & Wets, 1998, Theorem 1.25). The corresponding *proximal map* $\text{prox}_{\tau f}\colon \mathbb{R}^n \to \mathbb{R}^n$ is given by

$$\text{prox}_{\tau f}(\widehat{x}) := \arg\inf_x f(x) + \tfrac{1}{2\tau}\|x - \widehat{x}\|_2^2 = \widehat{x} - \tau\nabla\,\text{env}_{\tau f}(\widehat{x}) \tag{5}$$

and can be interpreted as a gradient descent step on the Moreau envelope. For a more detailed discussion of the connection between proximal map and Moreau evelope see e.g. Rockafellar & Wets (1998); Bauschke & Combettes (2011); Parikh & Boyd (2014).

The *proximal point method* (Rockafellar, 1976; Güler, 1992; Bauschke & Combettes, 2011; Parikh & Boyd, 2014) aims to minimize $f$ by iteratively updating $\widehat{x} \mapsto \text{prox}_{\tau f}(\widehat{x})$. Now, assume that $f$ decomposes as $f = g + h$, with $g$ differentiable and $h$ potentially non-smooth, and consider a linearization of $g$ around $\widehat{x}$ given by

$$\widetilde{g}(x) := g(\widehat{x}) + \langle x - \widehat{x}, \nabla g(\widehat{x})\rangle. \tag{6}$$

The corresponding proximal map reads as

$$\text{prox}_{\tau(\widetilde{g}+h)}(\widehat{x}) = \arg\inf_x \widetilde{g}(x) + h(x) + \tfrac{1}{2\tau}\|x - \widehat{x}\|_2^2 = \text{prox}_{\tau h}\big(\widehat{x} - \tau\nabla g(\widehat{x})\big) \tag{7}$$

and iterating $\widehat{x} \mapsto \text{prox}_{\tau h}\big(\widehat{x} - \tau\nabla g(\widehat{x})\big)$ is called *proximal gradient descent* (Nesterov, 1983; Combettes & Pesquet, 2011; Parikh & Boyd, 2014).

## 5 METHOD

Our goal is to translate the idea of proximal methods to parameterized optimization models as in Section 3, by defining a *Lagrange-Moreau envelope* of the loss $w \mapsto \ell(x^*(w))$ on which we can perform gradient descent. In analogy to (4), given $w$ and corresponding optimal solution $x^*$, the envelope should select an $x$ in the proximity of $x^*$ with a lower loss $\ell$. The key concept is to replace the Euclidean distance with a *Lagrangian divergence* indicating how close $x$ is to optimality given $w$.

### 5.1 LAGRANGIAN DIVERGENCE

First we define the *Lagrangian difference* for $x \in \mathcal{X}$ and $w \in \mathbb{R}^k$ as

$$D_{\mathcal{L}}(x, y|w) := \mathcal{L}(x, y, w) - \mathcal{L}^*(w), \tag{8}$$

where $\mathcal{L}^*(w)$ is the optimal Lagrangian (1). We then define the *Lagrangian divergence*[4] as

$$D_{\mathcal{L}}^*(x|w) := \sup_{y \in \mathcal{Y}} D_{\mathcal{L}}(x, y|w) = \sup_{y \in \mathcal{Y}}\big[\mathcal{L}(x, y, w) - \mathcal{L}^*(w)\big] \geq 0 \tag{9}$$

for $x \in \mathcal{X}$ and $w \in \mathbb{R}^k$, where the last inequality follows from the inequality

$$\sup_{y \in \mathcal{Y}} \mathcal{L}(x, y, w) \geq \inf_{\widetilde{x} \in \mathcal{X}} \sup_{y \in \mathcal{Y}} \mathcal{L}(\widetilde{x}, y, w) = \mathcal{L}^*(w). \tag{10}$$

The divergence has the key property

$$D_{\mathcal{L}}^*(x|w) = 0 \quad \text{if and only if} \quad x \in \mathcal{X} \text{ minimizes (2)} \tag{11}$$

which makes it a reasonable measure of optimality of $x$ given $w$, for the proof, see Appendix F.

---

[3] Our method only requires continuity of the solution mapping which a weaker assumption than differentiability. Therefore, whenever the true gradients exist, our continuity assumption is also fulfilled.

[4] In some cases, the Lagrangian divergence coincides with the *Bregman divergence*, a proximity measure generalizing the squared Euclidean distance, opening connections to *Mirror descent*, see Appendix D.

## 5.2 LAGRANGE-MOREAU ENVELOPE

Given $\tau > 0$, we say that $\ell_\tau : \mathbb{R}^k \to \mathbb{R}$ is the *lower Lagrange-Moreau envelope* ($\mathcal{L}$-envelope) if

$$\ell_\tau(w) := \inf_{x \in \mathcal{X}} \ell(x) + \tfrac{1}{\tau} D_\mathcal{L}^*(x|w) = \inf_{x \in \mathcal{X}} \sup_{y \in \mathcal{Y}} \ell(x) + \tfrac{1}{\tau} D_\mathcal{L}(x, y|w). \tag{12}$$

The corresponding *lower Lagrangian proximal map* $z_\tau : \mathbb{R}^k \to \mathbb{R}^{n+m}$ is defined as

$$z_\tau(w) := \arg \inf_{x \in \mathcal{X}} \sup_{y \in \mathcal{Y}} \ell(x) + \tfrac{1}{\tau} D_\mathcal{L}(x, y|w) = \arg \inf_{x \in \mathcal{X}} \sup_{y \in \mathcal{Y}} \mathcal{L}(x, y, w) + \tau \ell(x). \tag{13}$$

The *upper $\mathcal{L}$-envelope* $\ell^\tau : \mathbb{R}^k \to \mathbb{R}$ is defined with maximization instead of minimization as

$$\ell^\tau(w) := \sup_{x \in \mathcal{X}} \ell(x) - \tfrac{1}{\tau} D_\mathcal{L}^*(x|w) = \sup_{x \in \mathcal{X}} \inf_{y \in \mathcal{Y}} \ell(x) - \tfrac{1}{\tau} D_\mathcal{L}(x, y|w), \tag{14}$$

and the corresponding *upper $\mathcal{L}$-proximal map* $z^\tau : \mathbb{R}^k \to \mathbb{R}^{n+m}$ ($\mathcal{L}$-proximal map) as

$$z^\tau(w) := \arg \sup_{x \in \mathcal{X}} \inf_{y \in \mathcal{Y}} \ell(x) - \tfrac{1}{\tau} D_\mathcal{L}(x, y|w) = \arg \inf_{x \in \mathcal{X}} \sup_{y \in \mathcal{Y}} \mathcal{L}(x, y, w) - \tau \ell(x). \tag{15}$$

The lower and upper $\mathcal{L}$-envelope are lower and upper bound approximations of the loss $w \mapsto \ell(x^*(w))$, respectively. We emphasize that the solutions to (12) and (14) are in general set-valued and we assume that they are non-empty and admit a single-valued selection that is continuous at $w$, which we denote by (13) and (15). We will also work with the *average $\mathcal{L}$-envelope*

$$\ell_\tau(w) := \tfrac{1}{2} \big[ \ell_\tau(w) + \ell^\tau(w) \big]. \tag{16}$$

The different envelopes are closely related to right-, left- and double-sided directional derivatives.

## 5.3 LAGRANGIAN PROXIMAL POINT METHOD

Our goal will be to perform gradient descent on the $\mathcal{L}$-envelope (12, 14). By Oyama & Takenawa (2018, Proposition 4.1), the gradients of the $\mathcal{L}$-envelopes read

$$\nabla \ell_\tau(w) = \tfrac{1}{\tau} \nabla_w \big[ \mathcal{L}(w, z_\tau) - \mathcal{L}(z^*, w) \big] \quad \text{and} \quad \nabla \ell^\tau(w) = \tfrac{1}{\tau} \nabla_w \big[ \mathcal{L}(z^*, w) - \mathcal{L}(z^\tau, w) \big], \tag{17}$$

where we abbreviate $z_\tau = z_\tau(w)$ and $z^\tau = z^\tau(w)$. The proof is in Appendix F. In analogy to the proximal point method (5) we refer to GD using (17) as the *Lagrangian Proximal Point Method* (LPPM), or more specifically, LPPM$_\tau$, LPPM$\tau$ and LPPM$^\tau$ for GD on $\ell_\tau$, $\ell_\tau$ and $\ell^\tau$, respectively.

**Example 1** (Direct Loss Minimization)**.** For an input $\mu \in \mathbb{R}^p$, label $x_{\text{true}} \in \mathcal{X}$, loss $\ell : \mathcal{X} \times \mathcal{X} \to \mathbb{R}$, feature map $\Psi : \mathcal{X} \times \mathbb{R}^p \to \mathbb{R}^k$ and an optimization problem of the form

$$x^*(w, \mu) = \arg \min_{x \in \mathcal{X}} -\langle w, \Psi(x, \mu) \rangle \tag{18}$$

the LPPM$_\tau$ update (17) reads

$$\nabla \ell_\tau(w) = \tfrac{1}{\tau} \big[ \Psi(x^*, \mu) - \Psi(x_\tau, \mu) \big], \tag{19}$$

with $x^* = x^*(w, \mu)$ and

$$x_\tau = \arg \min_{x \in \mathcal{X}} -\langle w, \Psi(x, \mu) \rangle + \tau \ell(x, x_{\text{true}}). \tag{20}$$

This recovers the "towards-better" *Direct Loss Minimization* (DLM) update (McAllester et al., 2010), while the "away-from-worse" update corresponds to the LPPM$^\tau$ update, both of which were proposed in the context of taking the limit $\tau \to 0$ to compute the true gradients.

## 5.4 LAGRANGIAN PROXIMAL GRADIENT DESCENT

LPPM requires computing the $\mathcal{L}$-proximal map (13) or (15). This requires choosing and implementing an appropriate optimization algorithm, which due to the loss term might have a much higher complexity than the oracle used to solve the forward problem (2). Instead, we now aim to introduce an approximation of the loss that allows solving the $\mathcal{L}$-proximal map with the same solver oracle that was used on the forward pass. We first observe that in many cases the Lagrangian takes the form

$$\mathcal{L}(x, y, w) := \langle x, c \rangle + \Omega(x, y, v), \tag{21}$$

---

**Algorithm 1** Forward and Backward Pass of LPGD$_\tau$

---

**function** FORWARDPASS($w$)
  $z^* \leftarrow$ SolverOracle($w$)
    *// Solve optimization (2)*
  **save** $w$, $z^*$ for backward pass
  **return** $z^*$

**function** BACKWARDPASS($\nabla \ell = \nabla_z \ell(z^*), \tau$)
  **load** $(c, v) = w$ and $z^*$ from forward pass
  $w_\tau \leftarrow c + \tau \nabla \ell, v$        *// Perturb parameters*
  $\widetilde{z}_\tau \leftarrow$ SolverOracle($w_\tau$)   *// Solve (22), warmstart with $z^*$*
  $\nabla_w \widetilde{\ell}_\tau(w) = \frac{1}{\tau} \nabla_w \big[ \mathcal{L}(\widetilde{z}_\tau, w) - \mathcal{L}(z^*, w) \big]$     *// (24)*
  **return** $\nabla_w \widetilde{\ell}_\tau(w)$      *// Gradient of $\mathcal{L}$-envelope*

---

with linear parameters $c \in \mathbb{R}^n$, non-linear parameters $v \in \mathbb{R}^{k-n}$ and continuously differentiable $\Omega$. Our approximation, inspired by the proximal gradient descent (7), is to consider a linearization $\widetilde{\ell}$ of the loss $\ell$ at $x^*$.[5] Importantly, the loss linearization is only applied *after the solver* and does not approximate or linearize the solution mapping. Abbreviating $\nabla \ell = \nabla \ell(x^*)$, we get the $\mathcal{L}$-proximal maps

$$\widetilde{z}_\tau(w) := \arg\inf_{x \in \mathcal{X}} \sup_{y \in \mathcal{Y}} \langle x, c \rangle + \Omega(x, y, v) + \tau \langle x, \nabla \ell \rangle = z^*(c + \tau \nabla \ell, v), \tag{22}$$

$$\widetilde{z}^\tau(w) := \arg\inf_{x \in \mathcal{X}} \sup_{y \in \mathcal{Y}} \langle x, c \rangle + \Omega(x, y, v) - \tau \langle x, \nabla \ell \rangle = z^*(c - \tau \nabla \ell, v), \tag{23}$$

which can be computed with the same solver oracle used to solve the forward problem (2). Note that warm-starting the solver with $z^*$ can strongly accelerate the computation, often making the evaluation of the $\mathcal{L}$-proximal map much faster than the forward problem. This enables efficient computation of the $\mathcal{L}$-envelope gradient (17) as

$$\nabla \widetilde{\ell}_\tau(w) = \frac{1}{\tau} \nabla_w \big[ \mathcal{L}(w, \widetilde{z}_\tau) - \mathcal{L}(w, z^*) \big] \quad \text{and} \quad \nabla \widetilde{\ell}^\tau(w) = \frac{1}{\tau} \nabla_w \big[ \mathcal{L}(w, z^*) - \mathcal{L}(w, \widetilde{z}^\tau) \big]. \tag{24}$$

In analogy to proximal gradient descent (7) we refer to GD using (24) as *Lagrangian Proximal Gradient Descent* (LPGD), or more specifically, to LPGD$_\tau$, LPGD$\tau$ and LPGD$^\tau$ for GD on $\widetilde{\ell}_\tau$, $\widetilde{\ell}\tau$ and $\widetilde{\ell}^\tau$, respectively. LPGD smoothly integrates into existing automatic differentiation frameworks (Abadi et al., 2015; Bradbury et al., 2018; Paszke et al., 2019), by simply replacing the backward pass operation (co-derivative computation) as summarized for LPGD$_\tau$ in Algorithm 1.

**Example 2** (Blackbox Backpropagation). For a linear program (LP)[6]

$$x^*(c) = \arg\min_{x \in \mathcal{X}} \langle x, c \rangle, \tag{25}$$

the LPGD$_\tau$ update (24) reads

$$\nabla \widetilde{\ell}_\tau(c) = \frac{1}{\tau} \big[ \widetilde{x}_\tau(c) - x^*(c) \big] = \frac{1}{\tau} \big[ x^*(c + \tau \nabla \ell) - x^*(c) \big], \tag{26}$$

which recovers the update rule in *Blackbox Backpropagation* (BB) (Vlastelica et al., 2020). The piecewise affine interpolation of the loss $c \mapsto \widetilde{\ell}(x^*(c))$ derived in BB agrees with the lower $\mathcal{L}$-envelope $\widetilde{\ell}_\tau$.

**Example 3** (Implicit Differentiation by Perturbation). For a regularized linear program

$$x^*(c) = \arg\min_{x \in \mathcal{X}} \langle x, c \rangle + \Omega(x) \tag{27}$$

with a strongly convex regularizer $\Omega \colon \mathcal{X} \to \mathbb{R}$, the LPGD$\tau$ update (24) reads

$$\nabla \widetilde{\ell}\tau(c) = \frac{1}{2\tau} \big[ \widetilde{x}_\tau(c) - \widetilde{x}^\tau(c) \big] = \frac{1}{2\tau} \big[ x^*(c + \tau \nabla \ell) - x^*(c - \tau \nabla \ell) \big], \tag{28}$$

recovering the update in Domke (2010), where only the limit case $\tau \to 0$ is considered.

## 5.5   REGULARIZATION & AUGMENTED LAGRANGIAN

To increase the smoothness of the $\mathcal{L}$-envelope, we augment the Lagrangian with a strongly convex regularizer

$$\mathcal{L}_\rho(x, y, w) := \mathcal{L}(x, y, w) + \frac{1}{2\rho} \| x - x^* \|_2^2 \tag{29}$$

---

[5]Note that other approximations of the loss can also be used depending on the functional form of the Lagrangian. For example, for a Lagrangian with quadratic terms we could use a quadratic loss approximation.

[6]For an LP over a polytope $\mathcal{X}$ the space of possible solutions is discrete. Whenever the solution is unique, which is true for almost every $w$, the solution mapping is locally constant (and hence continuous) around $w$. Therefore our continuity assumptions hold for almost all $w$.

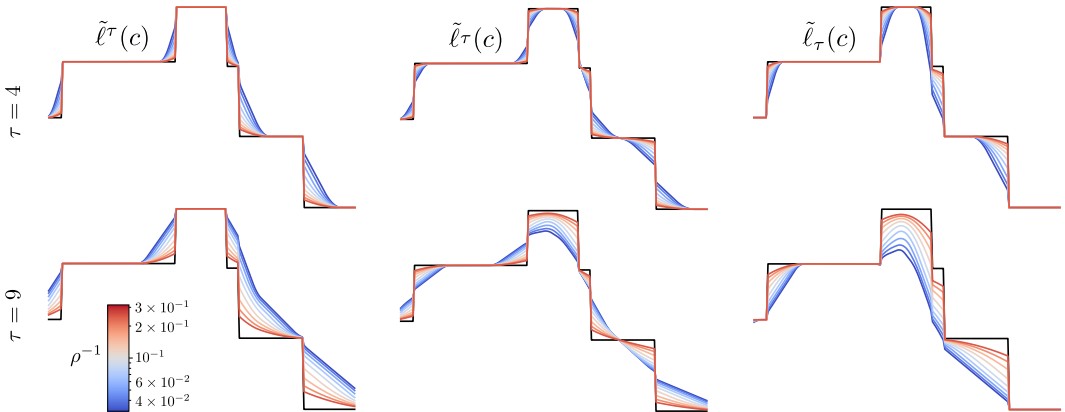

Figure 1: Visualization of the upper $\widetilde{\ell}^{\tau}$, average $\widetilde{\ell}_{\tau}$, and lower $\widetilde{\ell}_{\tau}$ Lagrange-Moreau envelope for different temperatures $\tau$ and augmentation strengths $\rho$. The envelopes are smoothed approximations of the linearized loss $c \mapsto \widehat{\ell}_{\tau}(x^*(c))$, illustrated in black. In Lagrangian Proximal Gradient Descent (LPGD) we optimize the loss by gradient descent on the Lagrange-Moreau envelope.

with $\rho > 0$. Equivalently, we may re-introduce the quadratic regularizer from the Moreau envelope (4) into the $\mathcal{L}$-envelope (12) and $\mathcal{L}$-proximal map (13)

$$\ell_{\tau\rho}(w) := \inf_{x \in \mathcal{X}} \sup_{y \in \mathcal{Y}} \ell(x) + \tfrac{1}{\tau} D_{\mathcal{L}}(x, y | w) + \tfrac{1}{2\rho} \|x - x^*\|_2^2, \tag{30}$$

$$z_{\tau\rho}(w) := \arg \inf_{x \in \mathcal{X}} \sup_{y \in \mathcal{Y}} \ell(x) + \tfrac{1}{\tau} D_{\mathcal{L}}(x, y | w) + \tfrac{1}{2\rho} \|x - x^*\|_2^2. \tag{31}$$

These definitions have an analogy for the upper envelope and for a linearized loss, which we omit for brevity. The $\text{LPPM}_\tau$ and $\text{LPGD}_\tau$ updates then take the form

$$\nabla \ell_{\tau\rho}(w) = \tfrac{1}{\tau} \nabla_w \big[ \mathcal{L}(w, z_{\tau\rho}) - \mathcal{L}(w, z^*) \big], \quad \nabla \widetilde{\ell}_{\tau\rho}(w) = \tfrac{1}{\tau} \nabla_w \big[ \mathcal{L}(w, \widetilde{z}_{\tau\rho}) - \mathcal{L}(w, z^*) \big]. \tag{32}$$

The augmentation does not alter the current optimal solution, but smoothens the Lagrange-Moreau envelope.[7] This also has connections to Jacobian-regularization in the implicit function theorem, which we discuss in Appendix C. Note that using this quadratic regularization with LPGD requires the solver oracle to support quadratic objectives, as is the case for the conic program solver used in our experiments. We visualize the smoothing of the different $\mathcal{L}$-envelopes of the linearized loss $c \mapsto \widetilde{\ell}(x^*(c))$ in Figure 1, for a quadratic loss on the solution to the linear program (25) with $\mathcal{X} = [0, 1]^n$ and a one-dimensional random cut through the cost space.

## 5.6 ASYMPTOTIC BEHAVIOR

We characterize the asymptotic behavior of the LPPM (17, 32) and LPGD (24, 32) updates. First, we consider the limit as $\tau \to 0$, in which the LPGD update is shown to converge to the true gradient.

**Theorem 5.1.** *Assume that $\mathcal{L} \in \mathcal{C}^2$ and assume that the solution mapping of optimization (2) admits a differentiable selection $x^*(w)$ at $w$. Then*

$$\lim_{\tau \to 0} \nabla \widetilde{\ell}_{\tau}(w) = \nabla_w \ell(x^*(w)) = \lim_{\tau \to 0} \nabla \widetilde{\ell}^{\tau}(w). \tag{33}$$

The proof, also highlighting the connections between $\mathcal{L}$-envelopes and directional derivatives, is given in Appendix F. Theorem 5.1 asserts that LPGD computes the same gradients in the limit as methods based on the implicit function theorem, such as *OptNet* (Amos & Kolter, 2017a), regularized LPs (Wilder et al., 2019a; Mandi & Guns, 2020), or differentiable conic programs (Agrawal et al., 2019a).

Next, we consider the limit $\tau \to \infty$. Let $\widehat{\mathcal{X}}(w)$ denote the *effective feasible set* defined as

$$\widehat{\mathcal{X}}(w) := \overline{\{x \in \mathcal{X} \mid \mathcal{D}_{\mathcal{L}}^*(x | w) < \infty\}}, \tag{34}$$

---

[7]Note that we ignore the dependence of $x^*$ on $w$ in the additional term, as it only serves the purpose of a regularizer and we do not aim to minimize $\|x - x^*\|_2$ directly.

where the bar denotes the closure. First, we have the result for the primal lower $\mathcal{L}$-proximal map (13).

**Proposition 5.2.** *Let $w$ be such that $\widehat{\mathcal{X}}(w)$ is nonempty. Then*

$$\lim_{\tau \to \infty} x_\tau(w) = \underset{x \in \widehat{\mathcal{X}}(w)}{\arg\inf} \ell(x) \tag{35}$$

*whenever the limit exists. For a linearized loss, we have*

$$\lim_{\tau \to \infty} \widetilde{x}_\tau(w) = \underset{x \in \widehat{\mathcal{X}}(w)}{\arg\inf} \langle x, \nabla \ell \rangle = x_{FW}(w), \tag{36}$$

*where $x_{FW}$ is the solution to a Frank-Wolfe iteration LP (Frank & Wolfe, 1956)*

Next proposition covers the case of the primal lower $\mathcal{L}$-proximal map (31) with a quadratic regularizer.

**Proposition 5.3.** *The primal lower $\mathcal{L}$-proximal map (31) turns into the standard proximal map (5)*

$$\lim_{\tau \to \infty} x_{\tau\rho}(w) = \underset{x \in \widehat{\mathcal{X}}(w)}{\arg\inf} \left[ \ell(x) + \tfrac{1}{2\rho} \|x - x^*\|_2^2 \right] = \operatorname{prox}_{\rho \ell + I_{\widehat{\mathcal{X}}(w)}}(x^*), \tag{37}$$

*whenever the limit exists. For a linearized loss, it reduces to the Euclidean projection onto $\widehat{\mathcal{X}}(w)$*

$$\lim_{\tau \to \infty} \widetilde{x}_{\tau\rho}(w) = \underset{x \in \widehat{\mathcal{X}}(w)}{\arg\inf} \left[ \langle x, \nabla \ell \rangle + \tfrac{1}{2\rho} \|x - x^*\|_2^2 \right] = P_{\widehat{\mathcal{X}}(w)}(x^* - \rho \nabla \ell). \tag{38}$$

The proofs can be found in Appendix F. The $\text{LPPM}_\tau$ (17, 32) and $\text{LPGD}_\tau$ (24, 32) updates corresponding to the $\mathcal{L}$-proximal maps (35, 37) and (36, 38) have the interpretation of decoupling the update step, by first computing a "target" (e.g. $x_{\tau\rho}$ via proximal gradient descent with step-size $\rho$), and then minimizing the Lagrangian divergence to make the target the new optimal solution.

We discuss multiple examples that showcase the asymptotic variations of LPPM and LPGD. Here, we will work with the finite-difference version of the updates (17, 32), which we denote by

$$\Delta \ell_\tau(w) \coloneqq \tau \nabla \ell_\tau(w), \qquad\qquad \Delta \ell^\tau(w) \coloneqq \tau \nabla \ell^\tau(w), \tag{39}$$

$$\Delta \ell_{\tau\rho}(w) \coloneqq \tau \nabla \ell_{\tau\rho}(w), \qquad\qquad \Delta \widetilde{\ell}_{\tau\rho}(w) \coloneqq \tau \nabla \widetilde{\ell}_{\tau\rho}(w). \tag{40}$$

**Example 4** (Identity with Projection). For an LP (25) it is $\widehat{\mathcal{X}}(w) = \mathcal{X}$ and we get the asymptotic regularized $\text{LPGD}_\tau$ update (32) in finite-difference form (40) as

$$\lim_{\tau \to \infty} \Delta \widetilde{\ell}_{\tau\rho}(c) = \lim_{\tau \to \infty} \left[ \widetilde{x}_{\tau\rho}(c) - x^* \right] = P_{\mathcal{X}}(x^* - \rho \nabla \ell) - x^*, \tag{41}$$

where we used (38). In the limit of large regularization $\rho \to 0$ with division by $\rho$ in analogy to Theorem 5.1, the above update converges to

$$\lim_{\rho \to 0} \lim_{\tau \to \infty} \tfrac{1}{\rho} \Delta \widetilde{\ell}_{\tau\rho}(c) = \lim_{\rho \to 0} \tfrac{1}{\rho} \left[ P_{\mathcal{X}}(x^* - \rho \nabla \ell) - x^* \right] = DP_{\mathcal{X}}(x^* | - \nabla \ell) = D^* P_{\mathcal{X}}(x^* | - \nabla \ell),$$

where $DP$ and $D^*P$ denote the directional derivative and coderivative of the projection $P$ at $x^*$. This is closely related to the *Identity with Projection* method by Sahoo et al. (2023), in which the true gradient is replaced by backpropagating $-\nabla \ell$ through the projection onto a relaxation of $\mathcal{X}$.[8]

**Example 5** (Smart Predict then Optimize). The *Smart Predict then Optimize* (SPO) setting (Mandi et al., 2020; Elmachtoub & Grigas, 2022) embeds an LP (25) as the final component of the prediction pipeline and assumes access to the ground truth cost $c_{\text{true}}$. The goal is to optimize the SPO loss $\ell_{\text{SPO}}(x^*(c), c_{\text{true}}) = \langle x^*(c) - x^*(c_{\text{true}}), c_{\text{true}} \rangle$. Due to the discreteness of the LP, the SPO loss has degenerate gradients with respect to $c$, i.e. they are zero almost everywhere and undefined otherwise. Choosing $\tau = \tfrac{1}{2}$ for the upper $\mathcal{L}$-proximal map (15), we get

$$x^{\frac{1}{2}}(c) = \underset{x \in \mathcal{X}}{\arg\max} \langle x - x^*(c_{\text{true}}), c_{\text{true}} \rangle - 2\langle x - x^*, c \rangle = \underset{x \in \mathcal{X}}{\arg\max} \langle x, c_{\text{true}} - 2c \rangle \tag{42}$$

---

[8]Note that this also has close ties to the one-step gradient arising in implicit differentiation of fixed-point iterations by treating the inverse Jacobian as an identity function (Geng et al., 2021; Chang et al., 2022; Bai et al., 2022). The projection operator arising in our setting is typically not present in these formulations as the variables are unconstrained, leaving only the identity as a replacement for the Jacobian.

which gives the lower and upper LPPM updates (17) in finite-difference form (39)

$$\Delta\ell_\tau(c) = x_\tau(c) - x^* \quad \text{and} \quad \Delta\ell^{\frac{1}{2}}(c) = x^* - x^{\frac{1}{2}}(c). \tag{43}$$

Summing both the updates and taking the limit $\tau \to \infty$ yields the combined LPPM update

$$\lim_{\tau\to\infty}\big[\Delta\ell_\tau(c) + \Delta\ell^{\frac{1}{2}}(c)\big] = \lim_{\tau\to\infty}\big[x_\tau(c) - x^{\frac{1}{2}}(c)\big] = x^*(c_{\text{true}}) - x^{\frac{1}{2}}(c) = \tfrac{1}{2}\nabla\ell_{\text{SPO+}}(c, c_{\text{true}}), \tag{44}$$

where we used (35). Note that as the SPO loss is already linear in $x$, LPPM and LPGD are equivalent. Update (44) recovers the gradient of the SPO+ loss

$$\ell_{\text{SPO+}}(c, c_{\text{true}}) := \sup_{x\in\mathcal{X}}\langle x, c_{\text{true}} - 2c\rangle + 2\langle x^*(c_{\text{true}}), c\rangle - \langle x^*(c_{\text{true}}), c_{\text{true}}\rangle \tag{45}$$

introduced by Elmachtoub & Grigas (2022), which has found widespread applications.

**Example 6** (Fenchel-Young Losses[9])**.** In the *structured prediction* setting we consider the regularized LP (27) as the final component of the prediction pipeline and assume access to the ground truth solutions $x_{\text{true}}$. The goal is to bring $x^*(c)$ close to $x_{\text{true}}$ by minimizing any loss $\ell(x)$ that is minimized over $\mathcal{X}$ at $x_{\text{true}}$. We compute the asymptotic $\text{LPPM}_\tau$ update (17) in finite-difference form (39) as

$$\lim_{\tau\to\infty}\Delta\ell_\tau(c) = \lim_{\tau\to\infty}\big[x_\tau(c) - x^*\big] = x_{\text{true}} - x^* = \nabla\ell_{\text{FY}}(c, x_{\text{true}}), \tag{46}$$

where we used (35) to compute the limit. This recovers the gradient of the Fenchel-Young loss[10]

$$\begin{aligned}
\ell_{\text{FY}}(c, x_{\text{true}}) &:= \max_{x\in\mathcal{X}}\big[\langle -c, x\rangle - \Omega(x)\big] + \Omega(x_{\text{true}}) - \langle -c, x_{\text{true}}\rangle \\
&= \langle c, x_{\text{true}}\rangle + \Omega(x_{\text{true}}) - \min_{x\in\mathcal{X}}\big[\langle c, x\rangle + \Omega(x)\big]
\end{aligned} \tag{47}$$

defined by Blondel et al. (2020). Depending on the regularizer $\Omega$ and the feasible region $\mathcal{X}$, Fenchel-Young losses cover multiple structured prediction setups, including the *structured hinge* (Tsochantaridis et al., 2005), *CRF* (Lafferty et al., 2001), and *SparseMAP* (Niculae et al., 2018) losses.

## 6 EXPERIMENTS

LPGD and LPPM are very useful tools for producing informative gradient replacements when the true gradient is degenerate. This has been repeatedly demonstrated by the numerous applications of the various special cases of LPGD and LPPM (Rolínek et al., 2020a;b; Mandi et al., 2020; Sahoo et al., 2023; Ferber et al., 2023). On the other hand, when the true gradient is non-degenerate, other variants of LPGD (Domke, 2010; McAllester et al., 2010) have been successfully applied to efficiently compute this gradient as $\tau \to 0$. However, our interpretation of LPGD computing more informative updates than normal gradients suggests that finite $\tau$ can be beneficial even in the latter case.

We explore this by comparing LPGD to gradient descent (GD) in an experimental setup in which non-degenerate gradients exist. To this end, we consider a version of the Sudoku experiment proposed by Amos & Kolter (2017a). The task is to learn the rules of Sudoku in the form of linear programming constraints from pairs of incomplete and solved Sudoku puzzles, see Appendix G for details.

We build on the CVXPY ecosystem (Diamond & Boyd, 2016; Agrawal et al., 2018; 2019a) to implement LPGD for a large class of parameterized optimization problems. CVXPY reduces parameterized optimization problems to conic programs in a differentiable way, which are then solved with the SCS solver (O'Donoghue et al., 2016) and differentiated based on Agrawal et al. (2019b).[11] As an alternative to the true derivative for the conic program, we implement Algorithm 1 (in all variations). This allows using LPGD for the large class of parameterized convex optimziation problems supported by CVXPY without modification.

The results of the Sudoku experiment are reported in Figure 2. $\text{LPGD}\tau$ reaches a lower final loss than GD, which shows that $\text{LPGD}\tau$ produces better update steps than standard gradients. $\text{LPGD}\tau$ also outperforms $\text{LPGD}^\tau$ and $\text{LPGD}_\tau$, which highlights that both lower and upper envelope carry relevant

---

[9]Note that an analogous derivation holds for *generalized* Fenchel-Young losses (Blondel et al., 2022), in which the regularized LP is replaced with a regularized energy function.

[10]The minus signs appear because Blondel et al. (2020) consider maximization instead of minimization.

[11]We modify this implementation to support the regularization term as described in Appendix C.

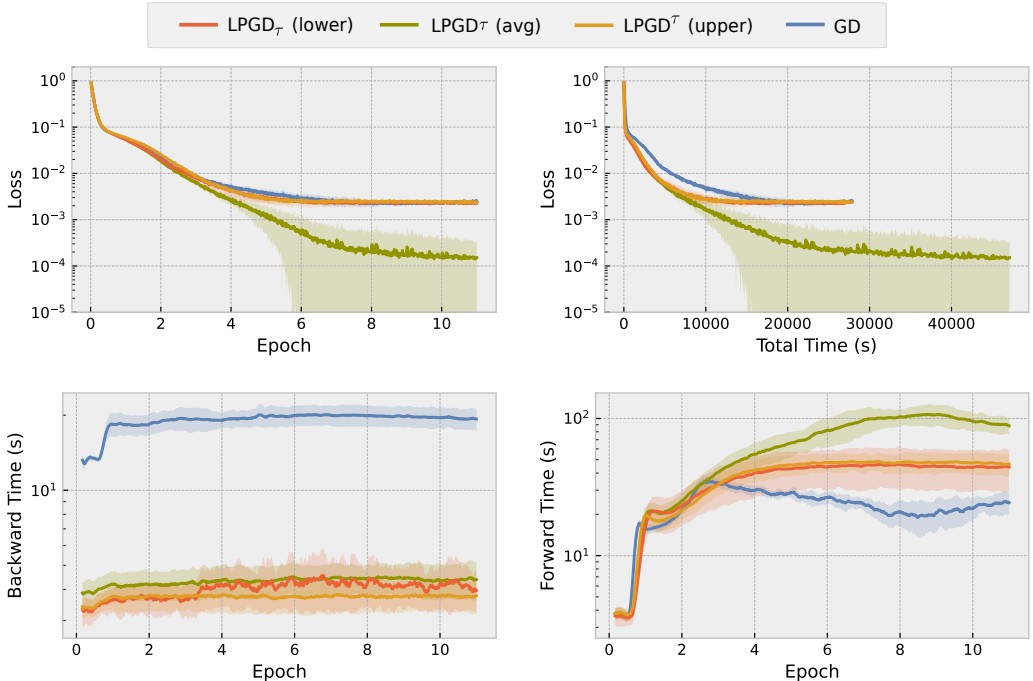

Figure 2: Comparison of LPGD and gradient descent (GD) on the Sudoku experiment. Reported is the train MSE over epochs and wall-clock time, as well as the time spent in the backward and forward passes. Statistics are over 5 restarts. Additional results can be found in Appendix G.

information for the optimization. This is intuitively understandable by considering that in Figure 1 $\text{LPGD}_\tau$ and $\text{LPGD}^\tau$ provide non-zero gradients in different subsets of the domain, while $\text{LPGD}\tau$ gives informative gradients in both subsets. We observe faster convergence of all variants of LPGD compared to GD in terms of wallclock time, which is due to the faster backward pass computation resulting from warmstarting. Note that the forward solution time over training increases especially for $\text{LPGD}\tau$, as the initial random optimization problem becomes more structured as the loss decreases.

## 7 CONCLUSION

We propose *Lagrangian Proximal Gradient Descent* (LPGD), a flexible framework for learning parameterized optimization models. **LPGD unifies and generalizes various state-of-the-art contemporary optimization methods**, including *Direct Loss Minimization* (McAllester et al., 2010), *Blackbox Backpropagation* (Vlastelica et al., 2020), *Implicit Differentiation by Perturbation* (Domke, 2010), *Identity with Projection* (Sahoo et al., 2023), *Smart Predict then Optimize* (Elmachtoub & Grigas, 2022), and *Fenchel-Young losses* (Blondel et al., 2020; 2022), and **provides deep links to traditional optimization methods**.

LPGD computes updates as finite-differences and only requires accessing the forward solver as a blackbox oracle, which makes it extremely simple to implement. We also provide an implementation of LPGD that smoothly integrates it into the CVXPY ecosystem (Diamond & Boyd, 2016; O'Donoghue et al., 2016; Agrawal et al., 2018; 2019b). Formulated as gradient descent on a loss function envelope, LPGD allows learning general objective and constraint parameters of saddlepoint problems even for solution mappings with degenerate derivatives.

Various special cases of LPGD have shown impressive results in optimizing parameters of solution mappings with degenerate derivatives and in speeding up computation of non-degenerate derivatives. We explore a new direction by using LPGD to efficiently compute informative updates even when non-degenerate derivatives exist. We find on a synthetic Sudoku experiment that LPGD achieves faster convergence and better final results when compared to gradient descent.

REPRODUCIBILITY STATEMENT

The experimental setup for the Sudoku experiment is described in detail in Appendix G. In addition, we will make the code for reproducing the experimental results as well as the LPGD implementation for CVXPY publicly available upon acceptance.

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

## A    LIMITATIONS

We are aware of few limitations of our method and are committed to transparent communication of them.

Reformulating the $\mathcal{L}$-proximal map as an instance of the forward solver makes the implementation simple and efficient. However, such reformulation is not possible in all cases since it requires access to the linear coefficients of the Lagrangian via the solver interface. This same issue arises for the augmentation introduced in Section 5.5, which requires accessing the quadratic coefficients of the solver and potentially turns an LP from the forward pass into a QP on the backward pass. This does not apply to our implementation of LPGD for the SCS solver (O'Donoghue et al., 2016) in CVXPY, as the solver natively supports quadratic conic programs.

When considering extremely small values of $\tau > 0$, the LPGD update requires solving the optimization problem very accurately, which can be expensive. Otherwise, warm-starting the solver on the backward pass with the forward pass solution will already satisfy the stopping criterion. However, we advocate for larger values of $\tau$ as the increased smoothing is usually beneficial. In our experiment, due to the convexity of the Lagrangian we are able to use the forward solver oracle to solve the optimization problems (2) and (31) to a very high accuracy in reasonable time using the SCS solver (O'Donoghue et al., 2016), and therefore did not observe issues arising from inexact solutions.

Finally, choosing the right combination of hyperparameters $\tau$ and $\rho$ can potentially require expensive tuning, as the different terms in the objective of the $\mathcal{L}$-envelope (32) can be of different magnitudes, depending on the problem at hand. A potential remedy is to normalize the terms onto a shared scale, which makes well-performing values of $\tau$ and $\rho$ more transferable and interpretable. Future work might also take inspiration from adaptive methods such as AIMLE (Minervini et al., 2023) for automatically tuning $\tau$ and $\rho$ on the fly.

## B    EXTENDED RELATED WORK

In this section we discuss the methods that are closest related to our framework.

**Direct Loss Minimization.**    In *Direct Loss Minimization* (McAllester et al., 2010) the goal is to directly optimize a structured prediction pipeline for a given task loss such as the BLEAU score, extending previous work using structured SVMs or CRFs. Given an input $\mu \in \mathbb{R}^p$ the prediction pipeline consists of a feature map $\Psi \colon \mathcal{X} \times \mathbb{R}^p \to \mathbb{R}^k \colon (x, \mu) \mapsto \Psi(x, \mu)$ and a corresponding parameterized Lagrangian (called score function) $\mathcal{L}(x, w) = \langle w, \Psi(x, \mu) \rangle$. The structured prediction is then computed by solving the embedded optimization problem

$$x^*(w, \mu) := \arg\max_{x \in \mathcal{X}} \langle w, \Psi(x, \mu) \rangle \tag{48}$$

over a finite set of solutions $\mathcal{X}$. Finally a task-specific loss $\ell \colon \mathcal{X} \times \mathcal{X} \to \mathbb{R}$ is used to compare the prediction to a label $x_{\text{true}} \in \mathcal{X}$. The goal is to optimize the loss over a dataset $\{(\mu_i, x_{\text{true},i})\}_{i=1}^N$, and the authors propose to optimize it using gradient descent. They show for the case $\mathcal{X} = \{-1, 1\}$ that the gradient can be computed using the limit of the finite difference

$$\nabla_w \ell(x^*(w, \mu)) = \pm \lim_{\tau \to 0} \tfrac{1}{\tau} \big[ \Psi(x_{\pm\tau}, \mu) - \Psi(x^*, \mu) \big] \rangle \tag{49}$$

with

$$x_{\pm\tau} := \arg\max_{x \in \mathcal{X}} \langle w, \Psi(x, \mu) \rangle \pm \tau \ell(x, x_{\text{true}}). \tag{50}$$

Where the two sign cases are called the "away-from-worse" and the "towards-better" update. The authors also discuss using relaxations of $\mathcal{X}$ as well has hidden variables that have similarities to our dual variables. The method is applied to phoneme-to-speech alignment. The DLM framework has also been generalized to non-linear objective functions (Song et al., 2016; Lorberbom et al., 2019), always considering the limit $\tau \to 0$, with applications to action classification, object detection, and semi-supervised learning of structured variational autoencoders.

**Blackbox Backpropagation.** In *Blackbox Backpropagation* (Vlastelica et al., 2020) the authors consider embedded combinatorial optimization problems with linear cost functions. Given a (potentially high-dimensional) input $\mu \in \mathbb{R}^p$ the prediction pipeline first computes the cost vector $c \in \mathbb{R}^n$ using a backbone model $W \colon \mathbb{R}^p \times \Theta \to \mathbb{R}^n \colon (\mu, \theta) \mapsto c$ and then solves the embedded linear optimization problem over a combinatorial space $\mathcal{X} \subset \mathbb{R}^n$

$$x^*(c) := \arg\min_{x \in \mathcal{X}} \langle x, c \rangle. \tag{51}$$

Finally the optimal solution is used as the prediction and compared to a label $x_{\text{true}} \in \mathcal{X}$ on a given loss $\ell \colon \mathcal{X} \times \mathcal{X} \to \mathbb{R}$. Because of the discrete solution space the gradient of the loss with respect to the cost vector (and therefore also the parameters $\theta$) is uninformative, as it is either zero or undefined. The authors propose to replace the uninformative gradient w.r.t. $c$ with the gradient of a piecewise-affine loss interpolation

$$\widetilde{\ell}_\tau(c) := \ell(x^*(c)) - \tfrac{1}{\tau} \min_{x \in \mathcal{X}} \langle c, x^*(c) - \widetilde{x}_\tau(c) \rangle \tag{52}$$

where we defined

$$\widetilde{x}_\tau(c) := \arg\min_{x \in \mathcal{X}} \langle x, c + \tau \nabla_x \ell(x^*) \rangle. \tag{53}$$

It has the gradient

$$\nabla \widetilde{\ell}_\tau(c) = -\tfrac{1}{\tau}[x^* - x^*(c + \tau \nabla_x \ell(x^*))]. \tag{54}$$

This approach has also been extended to learning discrete distributions in Niepert et al. (2021).

**Implicit Differentiation by Perturbation.** *Implicit Differentiation by Perturbation* (Domke, 2010) was proposed in the context of graphical models and marginal inference. The parameters $c$ now correspond to the parameters of an exponential family distribution and $\mathcal{X}$ is the marginal polytope. Computing the marginals requires solving the entropy-regularized linear program

$$\arg\max_{x \in \mathcal{X}} \langle x, c \rangle + \Omega(x), \tag{55}$$

where $\Omega$ denotes the entropy. The final loss function $\ell$ depends on the computed marginals and some data $x_{\text{true}}$. Approximate marginal inference can be seen as approximating the marginal polytope with a set of linear equality constraints

$$x^*(c) := \arg\max_{x \in \mathbb{R}^n, Ax=b} \langle x, c \rangle + \Omega(x), \tag{56}$$

which can be solved by loopy/tree-reweighted belief propagation. The authors show that in this case, as an alternative to implicit differentiation, the gradients of the loss with respect to the parameters can be computed by

$$\nabla_w \ell(x^*(w)) = \lim_{\tau \to 0} \tfrac{1}{\tau}[x^*(c + \tau \nabla_x \ell(x^*)) - x^*(c)]. \tag{57}$$

The method is applied to binary denoising, where the authors also test the double-sided perturbation case.

**Identity with Projection.** In *Identity with Projection* (Sahoo et al., 2023) the setup is the same as in Blackbox Backpropagation. The goal of this method is to speed up the backward pass computation by removing the second invocation of the solver oracle on the backward pass. In the basic version, the authors propose to replace the uninformative gradient through the solver $\nabla_c \ell(x^*(c))$ by simply treating the solver as a negative identity, and returning $\Delta_{\text{Id}} := -\nabla \ell(x^*)$ instead. Connections are drawn to the straight-through estimator. Further, the authors identify transformations of the cost vector $P \colon \mathbb{R}^k \to \mathbb{R}^k$ that leave the optimal solution unchanged (e.g. normalization), i.e. $x^*(P(c)) = x^*(c)$. They propose to refine the vanilla identity method by differentiating through the transformation, yielding the update $\Delta_{\text{Id}} := P'_{x^*}(-\nabla \ell(x^*))$. This is also shown to have an interpretation as differentiating through a projection onto a relaxation $\widetilde{\mathcal{X}}$ of the feasible space, i.e.

$$\Delta_{\text{Id}} = D^* P_{\widetilde{\mathcal{X}}}(x^*)(-\nabla \ell(x^*)). \tag{58}$$

Experimentally, the method is shown to be competitive with Blackbox Backpropagation and I-MLE.

**Smart Predict then Optimize.** In *Smart Predict then Optimize* (Elmachtoub & Grigas, 2022) setting the authors consider a linear program

$$x^*(c) := \arg\min_{x \in \mathcal{X}} \langle x, c \rangle \tag{59}$$

as the final component of a prediction pipeline. The cost parameters $c \in \mathbb{R}^n$ are not known with certainty at test time, and are instead predicted from an input $\mu \in \mathbb{R}^p$ via a prediction model $W_\theta \colon \mathbb{R}^p \to \mathbb{R}^n \colon \mu \mapsto c$ with parameters $\theta \in \Theta$. What distinguishes this setup from the one considered in e.g. Blackbox Backpropagation is that during training the true cost vectors $c_{\text{true}}$ are available, i.e. there is a dataset $\{\mu_i, c_{\text{true},i}\}_{i=1}^N$. A naive approach would be to directly regress the prediction model onto the true cost vectors by minimizing the mean squared error

$$\ell_{\text{MSE}}(\mu, c_{\text{true}}) = \tfrac{1}{2} \|W_\theta(\mu) - c_{\text{true}}\|. \tag{60}$$

However, the authors note that this ignores the actual downstream performance metric, which is the regret or SPO loss

$$\ell_{\text{SPO}}(x^*(c), c_{\text{true}}) = \langle x^*(c) - x^*(c_{\text{true}}), c_{\text{true}} \rangle. \tag{61}$$

Unfortunately this loss does not have informative gradients, so the authors propose to instead optimize a convex upper bound, the SPO+ loss

$$\ell_{\text{SPO+}}(c, c_{\text{true}}) := \sup_{x \in \mathcal{X}} \langle x, c_{\text{true}} - 2c \rangle + 2\langle x^*(c_{\text{true}}), c \rangle - \langle x^*(c_{\text{true}}), c_{\text{true}} \rangle, \tag{62}$$

a generalization of the hinge loss. Experimentally, results on shortest path problems and portfolio optimization demonstrate that optimizing the SPO+ loss offers significant benefits over optimizing the naive MSE loss.

**Fenchel-Young Losses.** *Fenchel Young Losses* (Blondel et al., 2020) were proposed in the context of structured prediction, in which the final prediction is the solution of a regularized linear program

$$x^*(c) := \arg\max_{x \in \mathcal{X}} \langle x, c \rangle - \Omega(x). \tag{63}$$

Supervision is assumed in the form of ground truth labels $x_{\text{true}}$. The authors propose to learn the parameters $c$ by minimizing the convex Fenchel Young Loss

$$\begin{aligned} \ell_{\text{FY}}(c, x_{\text{true}}) &:= \max_{x \in \mathcal{X}} \big[ \langle c, x \rangle - \Omega(x) \big] + \Omega(x_{\text{true}}) - \langle c, x_{\text{true}} \rangle \\ &= \langle c, x_{\text{true}} \rangle + \Omega(x_{\text{true}}) - \min_{x \in \mathcal{X}} \big[ \langle c, x \rangle + \Omega(x) \big], \end{aligned} \tag{64}$$

for which a variety of appealing theoretical results are presented. Many known loss functions from structured prediction and probabilistic prediction are shown to be recovered by Fenchel-Young losses for specific choices of feasible region $\mathcal{X}$ and regularizer $\Omega$, including the *structured hinge* (Tsochantaridis et al., 2005), *CRF* (Lafferty et al., 2001), and *SparseMAP* (Niculae et al., 2018) losses.

## C  IMPLICIT DIFFERENTIATION WITH AUGMENTATION

We inspect how the augmentation in (29) affects existing methods for computing the adjoint derivative of the augmented optimization problem

$$z_\rho^*(w) = (x_\rho^*(w), y_\rho^*(w)) := \arg\min_{x \in \mathcal{X}} \max_{y \in \mathcal{Y}} \mathcal{L}_\rho(x, y, w). \tag{65}$$

**Quadratic Program.** For a symmetric positive semi-definite matrix $H$ we can write a quadratic program with inequality constraints as

$$(x^*, s^*) = \arg\min_{x, s \geq 0} \tfrac{1}{2} x^T H x + c^T y \quad \text{subject to} \quad Ax + b + s = 0. \tag{66}$$

In Lagrangian form, we can write it as

$$z^* = (x^*, s^*, y^*) = \arg\min_{x, s \geq 0} \max_y \mathcal{L}(H, c, A, b, x, y) \tag{67}$$

with the Lagrangian

$$\mathcal{L}(x, s, y, H, A, b, c) = \tfrac{1}{2}x^T H x + c^T x + (Ax + b + s)^T y. \tag{68}$$

The augmentation in (29) augments the Lagrangian to

$$\mathcal{L}_\rho(x, s, y, H, A, b, c) = \tfrac{1}{2}x^T H x + c^T x + (Ax + b + s)^T y + \tfrac{1}{\rho}\|x - x^*\|_2^2 \tag{69}$$

and we write

$$z_\rho^*(H, A, b, c) = \arg \min_{x, s \geq 0} \max_y \mathcal{L}_\rho(x, s, y, H, A, b, c). \tag{70}$$

As described in (Amos & Kolter, 2017a), the optimization problem can be differentiated by treating it as an implicit layer via the KKT optimality conditions, which are given as

$$Hx + A^T y + c + \tfrac{2}{\rho}(x - x^*) = 0 \qquad \mathrm{diag}(y)s = 0 \qquad Ax + b + s = 0 \qquad s \geq 0. \tag{71}$$

Assuming strict complementary slackness renders the inequality redundant and the conditions reduce to the set of equations

$$0 = F_\rho(x, s, y, H, A, b, c) = \begin{pmatrix} Hx + A^T y + c + \tfrac{2}{\rho}(x - x^*) \\ \mathrm{diag}(y)s \\ Ax + b + s \end{pmatrix} \tag{72}$$

which admits the use of the implicit function theorem. It states that under the regularity condition that $\partial F_\rho / \partial z$ is invertible, $z_\rho^*(w)$ can be expressed as an implicit function, and we can compute its Jacobian by linearizing the optimality conditions around the current solution

$$0 = \frac{\partial F_\rho}{\partial z} \frac{\partial z_\rho^*}{\partial w} + \frac{\partial F_\rho}{\partial w} \tag{73}$$

with

$$\frac{\partial F_\rho}{\partial z} = \begin{bmatrix} H + \tfrac{2}{\rho}I & 0 & A^T \\ 0 & \mathrm{diag}(\tfrac{y}{s}) & I \\ A & I & 0 \end{bmatrix}. \tag{74}$$

It is now possible to compute the desired Vector-Jacobian-product as

$$\nabla_w \ell(x^*(w)) = \frac{\partial z^*}{\partial w}^T \nabla \ell(x^*) = -\frac{\partial F_\rho}{\partial w}^T \frac{\partial F_\rho}{\partial z}^{-T} \nabla \ell(x^*), \tag{75}$$

which involves solving a linear system. The augmentation term, therefore, serves as a regularizer for this linear system.

**Conic Program.** A conic program (Boyd & Vandenberghe, 2014) is defined as

$$(x^*, s^*) = \arg \min_{x, s \in \mathcal{K}} c^T y \quad \text{subject to} \quad Ax + b + s = 0 \tag{76}$$

where $\mathcal{K}$ is a cone. The Lagrangian of this optimization problem is

$$\mathcal{L}(x, s, y, A, b, c) = c^T x + (Ax + b + s)^T y \tag{77}$$

which allows an equivalent saddle point formulation given by

$$z^* = (x^*, s^*, y^*) = \arg \min_{x, s \in \mathcal{K}} \max_y \mathcal{L}(x, s, y, A, b, c). \tag{78}$$

The KKT optimality conditions are

$$A^T y + c = 0, \qquad Ax + s + b = 0, \qquad (s, y) \in \mathcal{K} \times \mathcal{K}^*, \qquad s^T y = 0, \tag{79}$$

where $\mathcal{K}^*$ is the dual cone of $\mathcal{K}$. The skew-symmetric mapping

$$Q(A, b, c) = \begin{bmatrix} 0 & A^T & c \\ -A & 0 & b \\ -c^T & -b^T & 0 \end{bmatrix} \tag{80}$$

is used in the homogenous self-dual embedding (O'Donoghue et al., 2016; Busseti et al., 2019), a feasibility problem that embeds the conic optimization problem. Agrawal et al. (2019b) solve and differentiate the self-dual embedding. We use the CVXPY implementation of this method as our baseline for computing the true adjoint derivatives of the optimization problem. The augmentation in (29) changes the stationarity condition and, thereby, the skew-symmetric mapping as

$$Q_\rho(A, b, c) = \begin{bmatrix} \tfrac{2}{\rho}I & A^T & c \\ -A & 0 & b \\ -c^T & -b^T & 0 \end{bmatrix}. \tag{81}$$

We adjust the CVXPY implementation accordingly for our experiments.

# D    RELATION TO MIRROR DESCENT

**Standard Mirror Descent.**    Classical mirror descent is an algorithm for minimizing a function $\ell(x)$ over a closed convex set $\mathcal{X} \subseteq \mathbb{R}^n$. The algorithm is defined by the distance-generating function (or mirror map) $\phi \colon \mathbb{R}^n \to \mathbb{R}$, a strictly convex continuously differentiable function. The mirror descent algorithm also requires the assumption that the dual space of $\phi$ is all of $\mathbb{R}^n$, i.e. $\{\nabla\phi(x) \mid x \in \mathbb{R}^n\} = \mathbb{R}^n$, and that the gradient of $\phi$ diverges as $\|x\|_2 \to \infty$.

The Bregman divergence of the mirror map is defined as

$$D_\phi(x, \widehat{x}) := \phi(x) - \phi(\widehat{x}) - \langle x - \widehat{x}, \nabla\phi(\widehat{x}) \rangle. \tag{82}$$

The lower[12] *Bregman-Moreau envelope* (Bauschke et al., 2018) $\mathrm{env}^\phi_{\tau f} \colon \mathbb{R}^n \to \mathbb{R}$ of a possibly non-smooth function $f \colon \mathbb{R}^n \to \mathbb{R}$ is defined for $\tau > 0$ as

$$\mathrm{env}^\phi_{\tau f}(\widehat{x}) := \min_x f(x) + \tfrac{1}{\tau} D_\phi(x, \widehat{x}). \tag{83}$$

The corresponding lower *Bregman-Moreau proximal map* $\mathrm{prox}^\phi_{\tau f} \colon \mathbb{R}^n \to \mathbb{R}^n$ is given by

$$\mathrm{prox}_{\tau f}(\widehat{x}) := \arg\inf_x f(x) + \tfrac{1}{\tau} D_\phi(x, \widehat{x}) \tag{84}$$

The superscript $\phi$ distinguishes the notation from the standard Moreau envelope (4) and proximal map (5).

Then, the mirror descent algorithm in proximal form is given by iteratively applying the Bregman-Moreau proximal map of $\tilde{\ell} + I_\mathcal{X}$ as

$$x_{k+1} = \arg\min_x \tilde{\ell}(x) + I_\mathcal{X}(x) + \tfrac{1}{\tau} D_\phi(x, x_k) \tag{85}$$

$$= \arg\min_{x \in \mathcal{X}} \langle x, \nabla\ell(x_k) \rangle + \tfrac{1}{\tau} D_\phi(x, x_k). \tag{86}$$

An equivalent form of this algorithm is

$$\eta_k = \nabla\phi(x_k) \tag{87}$$

$$\eta_{k+1} = \eta_k - \tau\nabla\ell(x_k) \tag{88}$$

$$\widetilde{x}_{k+1} = (\nabla\phi)^{-1}(\eta_{k+1}) \tag{89}$$

$$x_{k+1} = \arg\min_{x \in \mathcal{X}} D_\phi(x, \widetilde{x}_{k+1}) \tag{90}$$

which highlights that gradients are applied in the dual space of $\phi$, and $\nabla\phi$ serves as the mapping between primal and ("mirrored") dual space.

**Lagrangian Mirror Descent.**    In this section, we derive *Lagrangian Mirror Descent* (LMD), an alternative algorithm to LPGD inspired by mirror descent. We define

$$\mathcal{L}_w(x) := \sup_{y \in \mathcal{Y}} \mathcal{L}(x, y, w) \tag{91}$$

and assume that $\mathcal{L}_w$ is strongly convex and continuously differentiable in $x$. As the key step, we define the distance-generating function (mirror map) as $\phi = \mathcal{L}_w$. This has the interpretation that distances are measured in terms of the Lagrangian, similar to the intuition behind the previously defined Lagrangian divergence.

This mirror map leads to the Bregman divergence

$$D_{\mathcal{L}_w}(x, \widehat{x}) = \mathcal{L}_w(x) - \mathcal{L}_w(\widehat{x}) - \langle x - \widehat{x}, \nabla\mathcal{L}_w(\widehat{x}) \rangle \tag{92}$$

satisfying

$$D_{\mathcal{L}_w}(x, x^*) \geq 0, \tag{93}$$

$$D_{\mathcal{L}_w}(x, x^*) = 0 \Leftrightarrow x = x^*(w) \quad \text{for } x \in \mathcal{X}. \tag{94}$$

---

[12]The terms lower and upper are replaced with left and right in Bauschke et al. (2018).

We define the lower *Lagrange-Bregman-Moreau envelope* at $w$ as the Bregman-Moreau envelope at $x^* = x^*(w)$, i.e.

$$\ell_\tau^\phi(w) := \text{env}_{\tau\ell+I_\mathcal{X}}^{\mathcal{L}_w}(x^*) \tag{95}$$

$$= \min_x \ell(x) + I_\mathcal{X}(x) + \tfrac{1}{\tau} D_{\mathcal{L}_w}(x, x^*) \tag{96}$$

$$= \min_{x\in\mathcal{X}} \ell(x) + \tfrac{1}{\tau}(\mathcal{L}(x, w) - \mathcal{L}^*(w) - \langle x - x^*, \nabla_x\mathcal{L}(x^*, w)\rangle), \tag{97}$$

and the corresponding lower *Lagrange-Bregman-Moreau proximal map*

$$x_\tau^\phi(w) := \text{prox}_{\tau\ell+I_\mathcal{X}}^{\mathcal{L}_w}(x^*) \tag{98}$$

$$= \arg\min_x \ell(x) + I_\mathcal{X}(x) + \tfrac{1}{\tau} D_{\mathcal{L}_w}(x, x^*) \tag{99}$$

$$= \arg\min_{x\in\mathcal{X}} \ell(x) + \tfrac{1}{\tau}(\mathcal{L}(x, w) - \mathcal{L}^*(w) - \langle x - x^*, \nabla_x\mathcal{L}(x^*, w)\rangle). \tag{100}$$

Again, the superscript $\mathcal{L}$ distinguishes the notation from the lower Lagrange-Moreau envelope (12) and lower $\mathcal{L}$-proximal map (13).

Similar to how we defined LPGD as gradient descent on the Lagrange-Moreau envelope of the linearized loss, we define *Lagrangian Mirror Descent* (LMD) as gradient descent on the Lagrange-Bregman-Moreau envelope of the linearized loss, i.e.

$$\nabla_w\widetilde{\ell}_\tau^\phi(w) = \tfrac{1}{\tau}\nabla_w[\mathcal{L}(\widetilde{x}_\tau^\phi, w) - \mathcal{L}(x^*, w) - \langle\widetilde{x}_\tau^\phi - x^*, \nabla_x\mathcal{L}(x^*, w)\rangle]. \tag{101}$$

Again, the approximation allows efficiently computing the gradients using the forward solver as

$$\widetilde{x}_\tau^\phi(u, v) = \arg\inf_{x\in\mathcal{X}} \tau\langle x, \nabla\ell\rangle + \langle x, u\rangle + \Omega(x, y, v) - \langle x, \nabla_x\mathcal{L}(x^*, w)\rangle \tag{102}$$

$$= x^*(u + \tau\nabla\ell - \nabla_x\mathcal{L}(x^*, w), v). \tag{103}$$

If $\mathcal{X} = \mathbb{R}^n$, then the original optimization problem (2) is unconstrained and we have the optimality condition $\nabla_x\mathcal{L}(x^*, w) = 0$. Therefore, the Bregman divergence

$$D_{\mathcal{L}_w}(x, x^*) = \mathcal{L}_w(x) - \mathcal{L}_w(x^*) - \langle x - x^*, \nabla\mathcal{L}_w(x^*)\rangle \tag{104}$$

$$= \mathcal{L}(x, w) - \mathcal{L}(x^*, w) = \mathcal{L}(x, w) - \mathcal{L}^*(w) = D_\mathcal{L}(x|w) \tag{105}$$

is coincides with the Lagrangian divergence. It follows that, in this case, the Lagrange-Moreau envelope and LPGD coincide with the Lagrange-Bregman-Moreau envelope and LMD, respectively.

## E   EXTENSION TO GENERAL LOSS FUNCTIONS

**Loss on Dual Variables.**   In the main text we only considered losses depending only on the primal variables, i.e. $\ell(x)$. For a loss on the dual variables $\ell(y)$, if we assume strong duality of (1), we can reduce the situation to the primal case, as

$$y^*(w) = \arg\sup_{y\in\mathcal{Y}}\inf_{x\in\mathcal{X}} \mathcal{L}(x, y, w) = \arg\inf_{y\in\mathcal{Y}}\sup_{x\in\mathcal{X}} -\mathcal{L}(x, y, w). \tag{106}$$

This amounts to simply negating the Lagrangian in all equations while swapping $x$ and $y$.

**Loss on Primal and Dual Variables.**   The situation becomes more involved for a loss function $L(x, y)$ depending on both primal and dual variables. If it decomposes into a primal and dual component, i.e. $L(x, y) = \ell_p(x) + \ell_d(y)$, we can compute the envelopes of the individual losses independently. Note that a linearization of the loss $\widetilde{L}$ trivially decomposes this way. Adding the envelopes together yields a combined lower and upper envelope for the total loss as

$$(\ell_p)_\tau(w) + (\ell_d)_\tau(w) = \inf_{x\in\mathcal{X}}\sup_{y\in\mathcal{Y}}[\tfrac{1}{\tau}\mathcal{L}(x, y, w) + \ell_p(x)] - \sup_{y\in\mathcal{Y}}\inf_{x\in\mathcal{X}}[\tfrac{1}{\tau}\mathcal{L}(x, y, w) - \ell_d(y)], \tag{107}$$

$$(\ell_d)^\tau(w) + (\ell_p)^\tau(w) = \sup_{y\in\mathcal{Y}}\inf_{x\in\mathcal{X}}[\tfrac{1}{\tau}\mathcal{L}(x, y, w) + \ell_d(y)] - \inf_{x\in\mathcal{X}}\sup_{y\in\mathcal{Y}}[\tfrac{1}{\tau}\mathcal{L}(x, y, w) - \ell_p(x)]. \tag{108}$$

Assuming strong duality of these optimization problems, this leads to

$$(\ell_p)_\tau(w) + (\ell_d)_\tau(w) = \inf_{x\in\mathcal{X}}\sup_{y\in\mathcal{Y}}[\tfrac{1}{\tau}\mathcal{L}(x,y,w) + \ell_p(x)] - \inf_{x\in\mathcal{X}}\sup_{y\in\mathcal{Y}}[\tfrac{1}{\tau}\mathcal{L}(x,y,w) - \ell_d(y)], \quad (109)$$

$$(\ell_d)^\tau(w) + (\ell_p)^\tau(w) = \inf_{x\in\mathcal{X}}\sup_{y\in\mathcal{Y}}[\tfrac{1}{\tau}\mathcal{L}(x,y,w) + \ell_d(y)] - \inf_{x\in\mathcal{X}}\sup_{y\in\mathcal{Y}}[\tfrac{1}{\tau}\mathcal{L}(x,y,w) - \ell_p(x)]. \quad (110)$$

Strong duality holds in particular for a linearized loss $\tilde{L}$, as this only amounts to a linear perturbation of the original optimization problem. Unfortunately, computing the average envelope

$$(\ell_p)\tau(w) + (\ell_d)\tau(w) = \tfrac{1}{2}\Bigg\{ \inf_{x\in\mathcal{X}}\sup_{y\in\mathcal{Y}}[\tfrac{1}{\tau}\mathcal{L}(x,y,w) + \ell_p(x)] - \inf_{x\in\mathcal{X}}\sup_{y\in\mathcal{Y}}[\tfrac{1}{\tau}\mathcal{L}(x,y,w) - \ell_d(y)] \quad (111)$$

$$+ \inf_{x\in\mathcal{X}}\sup_{y\in\mathcal{Y}}[\tfrac{1}{\tau}\mathcal{L}(x,y,w) + \ell_d(y)] - \inf_{x\in\mathcal{X}}\sup_{y\in\mathcal{Y}}[\tfrac{1}{\tau}\mathcal{L}(x,y,w) - \ell_p(x)] \Bigg\}$$

$$(112)$$

or its gradient would now require four evaluations of the solver, which can be expensive. To reduce the number of evaluations, we instead "combine" the perturbations of the primal and dual loss, i.e. we define

$$L_\tau(w) := \inf_{x\in\mathcal{X}}\sup_{y\in\mathcal{Y}} L(x,y) + \tfrac{1}{\tau}[\mathcal{L}(x,y,w) - \mathcal{L}^*(w)] \quad (113)$$

$$L^\tau(w) := \inf_{x\in\mathcal{X}}\sup_{y\in\mathcal{Y}} L(x,y) + \tfrac{1}{\tau}[\mathcal{L}(x,y,w) - \mathcal{L}^*(w)] \quad (114)$$

$$L\tau(w) := \tfrac{1}{2}\{L_\tau(w) + L^\tau(w)\} \quad (115)$$

$$= \inf_{x\in\mathcal{X}}\sup_{y\in\mathcal{Y}}[\tfrac{1}{\tau}\mathcal{L}(x,y,w) + L(x,y)] - \inf_{x\in\mathcal{X}}\sup_{y\in\mathcal{Y}}[\tfrac{1}{\tau}\mathcal{L}(x,y,w) - L(x,y)]. \quad (116)$$

The above definitions are valid even when we do not have strong duality of (1) and (113). Note that $L_\tau$ and $L^\tau$ are not necessarily lower and upper bounds of the loss anymore. However, these combined envelopes also apply to loss functions that do not separate into primal and dual variables, and computing their gradients requires fewer additional solver evaluations. For $L(x,y) = \ell(x)$ we have

$$L_\tau(w) = \ell_\tau(w), \qquad L\tau(w) = \ell\tau(w), \qquad L^\tau(w) = \ell^\tau(w). \quad (117)$$

For the rest of the appendix, we will work with $L$ instead of $\ell$ for full generality and reduce the situation to a primal loss $\ell$ with the relations above.

# F  PROOFS

**Lemma F.1** (Equation (11)). *It holds that*

$$D^*_{\mathcal{L}}(x|w) = 0 \text{ if and only if } x \text{ minimizes (2) for } x \in \mathcal{X}. \quad (118)$$

*Proof of Lemma F.1.* If $x \in \mathcal{X}$ minimizes (2), this means

$$\sup_{y\in\mathcal{Y}}\mathcal{L}(x,y,w) = \inf_{\tilde{x}\in\mathcal{X}}\sup_{y\in\mathcal{Y}}\mathcal{L}(\tilde{x},y,w) \quad (119)$$

and therefore

$$D^*_{\mathcal{L}}(x|w) = \sup_{y\in\mathcal{Y}}\big[\mathcal{L}(x,y,w) - \mathcal{L}^*(w)\big] = \inf_{\tilde{x}\in\mathcal{X}}\sup_{y\in\mathcal{Y}}\big[\mathcal{L}(\tilde{x},y,w)\big] - \mathcal{L}^*(w) = 0. \quad (120)$$

If $D^*_{\mathcal{L}}(x|w) = 0$, we have from the definition of the Lagrangian divergence (9) that

$$\sup_{y\in\mathcal{Y}}\mathcal{L}(x,y,w) = \mathcal{L}^*(w) = \min_{\tilde{x}\in\mathcal{X}}\sup_{y\in\mathcal{Y}}\mathcal{L}(\tilde{x},y,w) \quad (121)$$

and hence $x$ is a minimizer of (2). $\qquad \square$

**Lemma F.2** (Equation (17)). *Assume that $\mathcal{L}, \ell \in \mathcal{C}^1$ and assume that the solution mappings of optimization (2, 12, 14) admit continuous selections $x^*(w), x_\tau(w), x^\tau(w)$ at $w$. Then*

$$\nabla \ell_\tau(w) = \tfrac{1}{\tau} \nabla_w \big[ \mathcal{L}(w, z_\tau) - \mathcal{L}(z^*, w) \big] \quad and \quad \nabla \ell^\tau(w) = \tfrac{1}{\tau} \nabla_w \big[ \mathcal{L}(z^*, w) - \mathcal{L}(z^\tau, w) \big]. \quad (122)$$

*Proof of Lemma F.2.* We assumed that $z^*(w)$ is a selection of the solution set continuous at $w$. We also assumed that $\mathcal{L}$ and $\ell$ are continuously differentiable. It then follows that

$$\nabla_w \ell_\tau(w) = \nabla_w \inf_{x \in \mathcal{X}} \sup_{y \in \mathcal{Y}} \ell(x) + \tfrac{1}{\tau} D_\mathcal{L}(x, y | w) \quad (123)$$

$$= \nabla_w \Big[ \inf_{x \in \mathcal{X}} \sup_{y \in \mathcal{Y}} [\ell(x) + \tfrac{1}{\tau} \mathcal{L}(x, y, w)] - \inf_{x \in \mathcal{X}} \sup_{y \in \mathcal{Y}} \tfrac{1}{\tau} \mathcal{L}(x, y, w) \Big] \quad (124)$$

$$= \nabla_w \Big[ \inf_{x \in \mathcal{X}} \sup_{y \in \mathcal{Y}} \ell(x) + \tfrac{1}{\tau} \mathcal{L}(x, y, w) \Big] - \nabla_w \Big[ \inf_{x \in \mathcal{X}} \sup_{y \in \mathcal{Y}} \tfrac{1}{\tau} \mathcal{L}(x, y, w) \Big] \quad (125)$$

$$= \nabla_w \big[ \ell(x_\tau) + \tfrac{1}{\tau} \mathcal{L}(x_\tau, y_\tau, w) \big] - \nabla_w \big[ \tfrac{1}{\tau} \mathcal{L}(x^*, y^*, w) \big] \quad (126)$$

$$= \tfrac{1}{\tau} \nabla_w \big[ \mathcal{L}(w, z_\tau) - \mathcal{L}(z^*, w) \big] \quad (127)$$

In the fourth equation we used the result by Oyama & Takenawa (2018, Proposition 4.1). The proof for the upper envelope is analogous. □

**Theorem 5.1.** *Assume that $\mathcal{L} \in \mathcal{C}^2$ and assume that the solution mapping of optimization (2) admits a differentiable selection $x^*(w)$ at $w$. Then*

$$\lim_{\tau \to 0} \nabla \widetilde{\ell}_\tau(w) = \nabla_w \ell(x^*(w)) = \lim_{\tau \to 0} \nabla \widetilde{\ell}^\tau(w). \quad (128)$$

*Proof of Theorem 5.1.* In this proof we work in the more general setup described in Appendix E. We aim to show that for a linear loss approximation $\widetilde{L}$, the LPGD update recovers the true gradient as $\tau$ approaches zero. We again assume the same form of the Lagrangian as in (21)

$$\mathcal{L}(z, w) = \langle z, u \rangle + \Omega(z, v) \quad (129)$$

with $w = (u, v)$ and get from Oyama & Takenawa (2018, Proposition 4.1)

$$\nabla_u \mathcal{L}^*(u, v) = \nabla_u \mathcal{L}(z^*, u, v) = z^*(u, v). \quad (130)$$

We define

$$\mathrm{d}w := \begin{pmatrix} \nabla_z L \\ 0 \end{pmatrix}. \quad (131)$$

Then it holds that

$$\lim_{\tau \to 0} \nabla_w \widetilde{L}_\tau(w) = \lim_{\tau \to 0} \tfrac{1}{\tau} [\nabla_w \mathcal{L}^*(w + \tau \mathrm{d}w) - \nabla_w \mathcal{L}^*(w)] \quad (132)$$

$$= \frac{\partial^2 \mathcal{L}^*}{\partial^2 w} \mathrm{d}w = \frac{\partial^2 \mathcal{L}^*}{\partial^2 w}^T \mathrm{d}w = \frac{\partial^2 \mathcal{L}^*}{\partial^2 w}^T \begin{pmatrix} \nabla_z L \\ 0 \end{pmatrix} \quad (133)$$

$$= \frac{\partial^2 \mathcal{L}^*}{\partial w \partial u}^T \nabla_z L = \frac{\partial (\nabla_u \mathcal{L}^*)}{\partial w}^T \nabla_z L = \frac{\partial z^*}{\partial w}^T \nabla_z L = \nabla_w L(z^*(w)). \quad (134)$$

The main step in this derivation was to identify the Jacobian of the solution mapping as a sub-matrix of the Hessian of the optimal Lagrangian function, which is a symmetric matrix under the conditions of Schwarz's theorem. A sufficient condition for this is the assumption that $\mathcal{L} \in \mathcal{C}^2$.[13] Exploiting the

---

[13]Note that a similar derivation already appeared in (Domke, 2010), but only for primal variables with linear parameters and without considering the benefits of finite values of $\tau$.

symmetry of the Hessian then allows computing the gradient, which is a co-derivative (backward-mode), as the finite-difference between two solver outputs, which usually only computes a directional (forward-mode) derivative as

$$\Delta z = \frac{\partial z^*}{\partial w}\Delta w = \lim_{\tau \to 0} \frac{1}{\tau}[z^*(w + \tau\Delta w) - z^*(w)] \tag{135}$$

$$= \lim_{\tau \to 0} \frac{1}{\tau}[\nabla_u \mathcal{L}^*(w + \tau\Delta w) - \nabla_u \mathcal{L}^*(w)]. \tag{136}$$

We observe that computing the gradient of the $\mathcal{L}$-envelope is the backward-mode counterpart of the forward-mode right-sided directional derivative. This observation also gives an interpretation of the LPGD update for finite $\tau$: In forward-mode, checking how the solver reacts to finite perturbation of the parameters intuitively provides higher-order information than linear sensitivities to infinitesimal perturbations. The finite-difference in the gradient of the $\mathcal{L}$-envelope serves the equivalent purpose in backward-mode, by back-propagating higher-order information instead of linear sensitivities as in standard directional co-derivatives (back-propagation).

Note that for $L(x, y) = \ell(x)$ this reduces to

$$\lim_{\tau \to 0} \nabla_w \tilde{\ell}_\tau(w) = \nabla_w \ell(x^*(w)). \tag{137}$$

An analogous proof and discussion also hold for the upper envelope $\tilde{\ell}^\tau$ and average envelope $\tilde{\ell}_\tau$, corresponding to the left- and double-sided directional derivatives. $\qquad\square$

**Proposition 5.2.** *Assume $\ell$ is continuous and finite-valued on $\mathcal{X}$. Let $w$ be a parameter for which*

$$\widehat{\mathcal{X}}(w) \coloneqq \overline{\{x \in \mathcal{X} \mid \mathcal{D}^*_{\mathcal{L}}(x|w) < \infty\}} \tag{138}$$

*is nonempty. Then*

$$\lim_{\tau \to \infty} x_\tau(w) = \underset{x \in \widehat{\mathcal{X}}(w)}{\arg\min} \ell(x) \tag{139}$$

*whenever the limit exists. For a linearized loss, we have*

$$\lim_{\tau \to \infty} \tilde{x}_\tau(w) = \underset{x \in \widehat{\mathcal{X}}(w)}{\arg\inf} \langle x, \nabla\ell \rangle = x_{FW}(w), \tag{140}$$

*where $x_{FW}$ is the solution to a Frank-Wolfe iteration LP (Frank & Wolfe, 1956)*

*Proof of Proposition 5.2.* Throughout the proof, let $w$ be a fixed parameter. For $\tau > 0$ we define $f_\tau \colon \mathcal{X} \to \mathbb{R}$ by

$$f_\tau(x) = \ell(x) + \frac{1}{\tau} D^*_{\mathcal{L}}(x|\tau) \quad \text{for } x \in \mathcal{X}. \tag{141}$$

Since $D^*_{\mathcal{L}} \geq 0$ on $\mathcal{X}$, we have that

$$f_\tau \geq f_{\tau'} \geq \ell \quad \text{on } \mathcal{X} \tag{142}$$

whenever $\tau' \geq \tau$. Now let $(x_\tau)_{\tau>0}$ be minimizers of (141) such that $x_\tau \to x \in \mathcal{X}$ as $\tau \to \infty$. We show that $x$ minimizes $\ell(x)$ over $\widehat{\mathcal{X}}(w)$.

To this end, let $(\tau_n)_{n=1}^\infty$ be a non-decreasing sequence such that $\tau_n \to \infty$ and denote $x_n = x_{\tau_n}$ and $f_n = f_{\tau_n}$, for short. Since $\widehat{\mathcal{X}}(w)$ is nonempty and $\ell$ is finite-valued on $\mathcal{X}$, we have that $f_n(x_n) < \infty$ and $x_n \in \widehat{\mathcal{X}}(w)$ for all $x \in \mathbb{N}$. Since $\widehat{\mathcal{X}}(w)$ is closed, also $x \in \widehat{\mathcal{X}}(w)$. Next, by (142), the sequence $f_n(x_n)$ for $n \in \mathbb{N}$ is nonincreasing and bounded from below (by a minimum of $\ell$ on $\{x_n \mid n \in \mathbb{N}\}$). Therefore, $f_n(x_n)$ is convergent. By the continuity of $\ell$, we have that $\ell(x_n) \to \ell(x)$ and thus $\frac{1}{\tau_n} D^*_{\mathcal{L}}(x_n|w)$ converges to some $c \geq 0$. Altogether, for any $\hat{x} \in \widehat{\mathcal{X}}(w)$, we have

$$\ell(x) + c = \lim_{n \to \infty} f_n(x_n) = \lim_{n \to \infty} \ell(x_n) + \frac{1}{\tau_n} D^*_{\mathcal{L}}(x_n|w) \leq \lim_{n \to \infty} \ell(\hat{x}) + \frac{1}{\tau_n} D^*_{\mathcal{L}}(\hat{x}|w) = \ell(\hat{x}).$$

The particular choice $\hat{x} = x$ shows that $c = 0$. Therefore $\ell(x) \leq \ell(\hat{x})$ for any $\hat{x} \in \widehat{\mathcal{X}}(w)$ as desired. The second part of the proposition follows directly by taking a loss linearization. $\qquad\square$

**Proposition 5.3.** *Assume $\ell$ is continuous and finite-valued on $\mathcal{X}$. Let $w$ be a parameter for which $\widehat{\mathcal{X}}(w)$ is nonempty. The primal lower $\mathcal{L}$-proximal map (31) turns into the standard proximal map (5)*

$$\lim_{\tau \to \infty} x_{\tau\rho}(w) = \arg\inf_{x \in \widehat{\mathcal{X}}(w)} \left[\ell(x) + \tfrac{1}{2\rho}\|x - x^*\|_2^2\right] = \operatorname{prox}_{\rho\ell + I_{\widehat{\mathcal{X}}(w)}}(x^*), \tag{143}$$

*whenever the limit exists. For a linearized loss, it reduces to the Euclidean projection onto $\widehat{\mathcal{X}}(w)$*

$$\lim_{\tau \to \infty} \widetilde{x}_{\tau\rho}(w) = \arg\inf_{x \in \widehat{\mathcal{X}}(w)} \left[\langle x, \nabla\ell\rangle + \tfrac{1}{2\rho}\|x - x^*\|_2^2\right] = P_{\widehat{\mathcal{X}}(w)}(x^* - \rho\nabla\ell). \tag{144}$$

*Proof of Proposition 5.3.* The proof is analogous to that of Proposition 5.2 with

$$f_\tau \colon \mathcal{X} \to \mathbb{R} \colon x \mapsto \ell(x) + \tfrac{1}{\tau}\mathcal{D}_{\mathcal{L}}^*(x|w) + \tfrac{1}{2\rho}\|x - x^*\|_2^2 \tag{145}$$

$\square$

## G  EXPERIMENTS

Instead of the mini-Sudoku case ($4 \times 4$ grid) in Amos & Kolter (2017a), we consider the full $9 \times 9$ Sudoku grid. The Sudoku board is modelled as a one-hot-encoding, with incomplete input and solved label $x_{\text{inc}}, x_{\text{true}} \in \{0, 1\}^{9 \times 9 \times 9}$. The optimization problem is modelled as a generic box-constrained linear program

$$x^*(A, b; x_{\text{inc}}) = \arg\min_{x \in \mathcal{X}} \langle x, x_{\text{inc}}\rangle \quad \text{subject to} \quad Ax + b = 0 \tag{146}$$

with $\mathcal{X} = [0, 1]^{9 \times 9 \times 9}$ and a sufficient number of constraints $m$ to represent the rules of the LP Sudoku.[14] In saddle-point formulation the optimization problem is

$$z^*(A, b; x_{\text{inc}}) = \arg\inf_{x \in \mathcal{X}} \sup_{y \in \mathbb{R}^m} \mathcal{L}(x, y, A, b, x_{\text{inc}}), \tag{147}$$

with $\mathcal{L}(x, y, A, b, c) = \langle x, c\rangle + \langle y, Ax + b\rangle$. Note that we have the effective feasible set

$$\widehat{\mathcal{X}}(A, b) = \{x \in [0, 1]^{9 \times 9 \times 9} \mid Ax + b = 0\}. \tag{148}$$

We follow the training protocol described by Amos & Kolter (2017a) that minimizes the mean square error between predictions $x^*(A, b; x_{\text{inc}})$ and one-hot encodings of the correctly solved Sudokus $x_{\text{true}}$. For evaluation, we follow Amos & Kolter (2017a) in refining the predictions by taking an argmax over the one-hot dimension and report the percentage of violated ground-truth Sudoku constraints as the error.

We modify the public codebase from Amos & Kolter (2017b). The dataset consists of 9000 training and 1000 test instances. We choose the hyperparameters learning rate $\alpha$, $\tau$ and $\rho$ with a grid search. The best hyperparameters for LPGD are $\tau = 10^4$, $\rho = 0.1$, $\alpha = 0.1$, for gradient descent they are $\rho = 10^{-3}$, $\alpha = 0.1$. We use these hyperparameters in our evaluation.

Additional results, including test losses and errors, are reported in Figure 3 and 4. We observe that there is not significant difference between train and test metrics, which shows that our formulation of Sudoku (146) allows generalizing across instances.

---

[14]The formulation as an LP differs from the original formulation by Amos & Kolter (2017a), in which a quadratic regularizer has to be added to meet the method requirements.

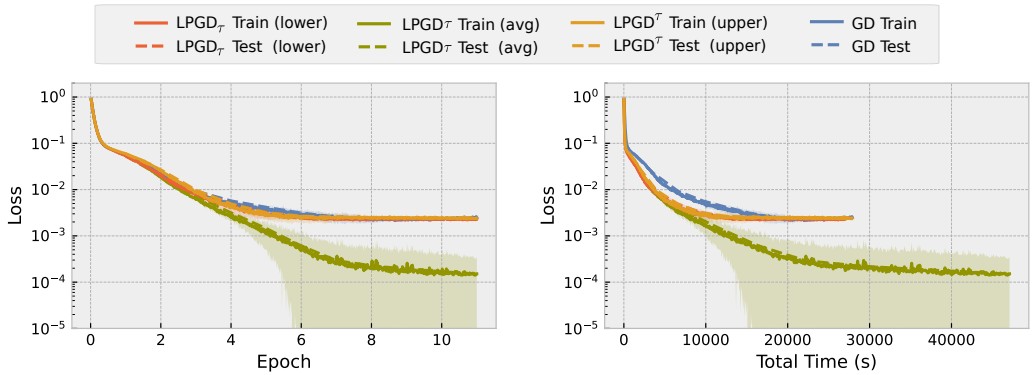

Figure 3: Comparison of LPGD and gradient descent (GD) on the Sudoku experiment. Reported are train and test MSE loss over epochs and wall-clock time. Statistics are over 5 restarts.

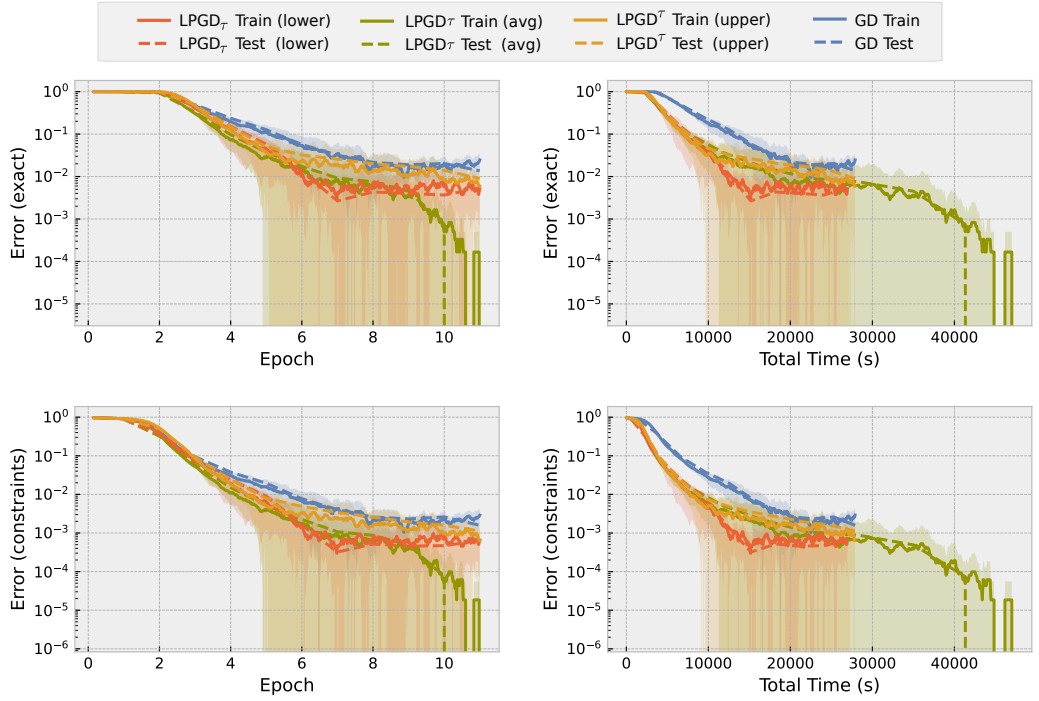

Figure 4: Comparison of LPGD and gradient descent (GD) on the Sudoku experiment. Reported are train and test errors over epochs and wall-clock time. The exact error refers to the proportion of incorrect solutions (at least one violated Sudoku constraint), while the constraint error refers to the proportion of violated Sudoku constraints. Statistics are over 5 restarts.

# H LIST OF SYMBOLS

| | |
|---|---|
| $x$ | primal variables |
| $s$ | optional slack variables |
| $y$ | dual variables |
| $z = (x, y)$ | primal & dual variables |
| $z^* = (x^*, y^*)$ | optimal primal & dual variables (2) |
| $\mathcal{X}, \mathcal{Y}$ | primal and dual feasible sets |
| $\widetilde{\mathcal{X}}$ | primal effective feasible set (34) |
| $w$ | all parameters |
| $u$ | linear parameters |
| $v$ | non-linear parameters |
| $c$ | linear primal parameters |
| $b$ | linear dual parameters |
| $\mathcal{L}$ | Lagrangian |
| $\mathcal{L}^*$ | optimal Lagrangian (1) |
| $\Omega$ | non-linear part of Lagrangian |
| $\tau$ | perturbation strength parameter |
| $\mathrm{env}_{\tau f}$ | Moreau envelope of a function $f$ (4) |
| $\mathrm{prox}_{\tau f}$ | Proximal map (5) |
| $\ell$ | loss on primal variables |
| $D_{\mathcal{L}}$ | Lagrangian difference (8) |
| $D_{\mathcal{L}}^*$ | Lagrangian divergence (9) |
| $\ell_\tau, \ell^\tau, \ell\tau$ | Lagrange-Moreau envelopes (12, 14, 16) |
| $z_\tau = (x_\tau, y_\tau)$ | lower $\mathcal{L}$-proximal map (13) |
| $z^\tau = (x^\tau, y^\tau)$ | upper $\mathcal{L}$-proximal map (15) |
| $\nabla\ell_\tau, \nabla\ell^\tau, \nabla\ell\tau$ | LPPM updates (17) |
| $\Delta\ell_\tau, \Delta\ell^\tau, \Delta\ell\tau$ | LPPM finite-differences (39) |
| $\widetilde{\ell}$ | linearization of $\ell$ at $x^*$ (6) |
| $\widetilde{\ell}_\tau, \widetilde{\ell}^\tau, \widetilde{\ell}\tau$ | Lagrange-Moreau envelopes of a linearized loss $\widetilde{\ell}$ |
| $\widetilde{z}_\tau = (\widetilde{x}_\tau, \widetilde{y}_\tau)$ | corresponding lower $\mathcal{L}$-proximal map (22) |
| $\widetilde{z}^\tau = (\widetilde{x}^\tau, \widetilde{y}^\tau)$ | corresponding upper $\mathcal{L}$-proximal map (23) |
| $\nabla\widetilde{\ell}_\tau, \nabla\widetilde{\ell}^\tau, \nabla\widetilde{\ell}\tau$ | LPGD updates (24) |
| $\Delta\widetilde{\ell}_\tau, \Delta\widetilde{\ell}^\tau, \Delta\widetilde{\ell}\tau$ | LPGD finite-differences |
| $\rho$ | augmentation strength parameter |
| $\mathcal{L}_\rho$ | augmented Lagrangian (29) |
| $\ell_{\tau\rho}, \ell^{\tau\rho}, \ell\tau\rho$ | augmented Lagrange-Moreau envelopes (30) |
| $z_{\tau\rho} = (x_{\tau\rho}, y_{\tau\rho})$ | augmented lower $\mathcal{L}$-proximal map (31) |
| $z^{\tau\rho} = (x^{\tau\rho}, y^{\tau\rho})$ | augmented upper $\mathcal{L}$-proximal map |
| $\nabla\ell_{\tau\rho}, \nabla\ell^{\tau\rho}, \nabla\ell\tau\rho$ | augmented LPPM updates (32) |
| $\Delta\ell_{\tau\rho}, \Delta\ell^{\tau\rho}, \Delta\ell\tau\rho$ | augmented LPPM finite-differences (40) |
| $\widetilde{\ell}_{\tau\rho}, \widetilde{\ell}^{\tau\rho}, \widetilde{\ell}\tau\rho$ | augmented Lagrange-Moreau envelopes of a linearized loss $\widetilde{\ell}$ |
| $\widetilde{z}_{\tau\rho} = (\widetilde{x}_{\tau\rho}, \widetilde{y}_{\tau\rho})$ | corresponding augmented lower $\mathcal{L}$-proximal map |
| $\widetilde{z}^{\tau\rho} = (\widetilde{x}^{\tau\rho}, \widetilde{y}^{\tau\rho})$ | corresponding augmented upper $\mathcal{L}$-proximal map |
| $\nabla\widetilde{\ell}_{\tau\rho}, \nabla\widetilde{\ell}^{\tau\rho}, \nabla\widetilde{\ell}\tau\rho$ | augmented LPGD updates (32) |
| $\Delta\widetilde{\ell}_{\tau\rho}, \Delta\widetilde{\ell}^{\tau\rho}, \Delta\widetilde{\ell}\tau\rho$ | augmented LPGD finite-differences (40) |
| $L$ | loss on primal & dual variables |
| $\widetilde{L}$ | linearization of $L$ at $z^*$ |

