# OpenReview forum: "Lagrangian Proximal Gradient Descent for Learning Convex Optimization Models"
_ICLR.cc/2024/Conference — Submitted to ICLR 2024_

### Official Review · Reviewer_t8qX · 2023-10-28

**Soundness:** 3 good
**Presentation:** 3 good
**Contribution:** 3 good
**Rating:** 6
**Confidence:** 4

**Summary:**

The paper proposes Lagrangian Proximal Gradient Descent (LPGD), an optimization framework for learning convex optimization models. It is effective for problems with non-differentiable loss functions and can handle models where gradients are not readily available. LPGD smooths the loss function, making optimization feasible in such scenarios. It efficiently computes updates by rerunning the forward solver and converges to the true gradient as the smoothing parameter approaches zero. LPGD can also offer benefits in fully different settings, making it a versatile tool for various optimization tasks. The method is interesting with promising performance.

**Strengths:**

The authors provide a new method, which considers a model with an embedded constrained convex optimization. They also provide a new update, the Lagrangian divergence proximal operator, which generalizes the classical Bregman divergences. Compared to classical proximal gradient descent methods, the update is new.

**Weaknesses:**

1. The proposed method does not come with the convergence rate analysis.

2. Could the authors provide examples of neural network-type constrained minimization problems? The neural network can be chosen as a simple one. This is to demonstrate the difference and the connection between current methods and the classical ones.

**Questions:**

One expects to see the method working on a neural network-type constrained optimization problem.

---

> ### Author Response · Authors · 2023-11-21
> **Reply to Reviewer t8qX**
>
> We thank the reviewer for the helpful feedback. We reply to the individual points below.
>
> - *The proposed method does not come with the convergence rate analysis.*
>     - It is very challenging even for the special cases to derive convergence rates, which is reflected by only the SPO+ and the Fenchel-Young loss in example 5 and 6 coming with convergence rates. For these methods it is possible because the loss is assumed to be convex in the parameters. In our generalized setup this is not necessarily true.
>     - We are also mostly interested in the case in which the model $W_\theta$ is a neural network, which means that reasonable convergence results are in general hard to provide.
>     - However, we agree that deriving convergence results for out algorithm under stronger assumptions would be a very nice addition to our work. Especially because due to our unification of previous methods, any convergence results would also directly apply to multiple existing methods. In this paper we focus on the first step of showing the similarities between various seemingly disconnected methods, and we leave the challenging analysis of the convergence to future work.
>
> - *Could the authors provide examples of neural network-type constrained minimization problems? The neural network can be chosen as a simple one. This is to demonstrate the difference and the connection between current methods and the classical ones.* and *One expects to see the method working on a neural network-type constrained optimization problem*
>     - Many of the methods that are covered as special cases have already been applied to problems involving neural networks. Examples include extracting edge costs in a shortest path problem from visual input (example 2 & 4, Vlastelica2020), extracting edge costs in graph matching problems for keypoint matching from visual input (example 2 & 4, Rolinek2020a) or extracting rankings in a retrieval system from visual input (example 2 & 4, Rolinek2020b). We added more detailed information on the experiments that have been proposed in the preceding literature of methods that we generalize.
>     - However, we want to highlight that the distinguishing difference between our proposed method and the methods which we refer to as traditional or classical, is that we consider the bilevel optimization problem instead of single-level optimization problems for which proximal methods were originally designed. We clarify this in the revision by explicitly formulating the bilevel problem in eq. 3.

---

> > ### Comment · Reviewer_t8qX · 2023-11-21
> > **Review to rebuttal**
> >
> > Thanks for addressing my questions. I read the rebuttal and will keep my score.

---

### Official Review · Reviewer_n99L · 2023-10-28

**Soundness:** 3 good
**Presentation:** 3 good
**Contribution:** 2 fair
**Rating:** 8
**Confidence:** 3

**Summary:**

This paper proposes a class of Lagrange Proximal Gradient Descent method. As the authors claimed, the main contribution of this work is that It generalizes many existing methods. The technical part of this paper is solid, and the paper is well-written and easy to follow.

**Strengths:**

I read over the paper. The technical part of this paper is solid and the ideas are promising. Similar to the idea of Mirror descent, they consider a specific distance function (Eq (11)). This idea is not surprising to me, but the authors then construct lower and upper Lagrange-Moreau envelop, and average them to improve the accuracy of the gradient with respect to the parameter w. This idea is similar to the idea of the central difference method which usually has higher precision than the forward and the backward difference method. They also demonstrated this by numerical experiments.

**Weaknesses:**

1. The literature survey in this paper is not good enough. They surveyed many papers in Section 2, but the details of the most relevant papers are not provided either in the main text or in the Appendix. I'm particularly interested in existing papers that can also compute the (approximate) gradient of the function $\ell(x^\star(w))$. Moreover, the authors claim that the proposed method generalizes many existing methods and discuss this in Example 1,2,3. For more general problems rather than these examples, whether the proposed method generalize existing methods?

2. From the experiment (Figure 2), even in terms of Epochs, LPGD-medium $\tau$ is faster than GD. This is not straightforward to me. If I understand correctly, all the simulated methods in the experiments are based on GD, while the gradient is computed based on different approaches. Therefore, I wonder whether the poor performance of GD comes from the low accuracy of gradient information computed by the method in Argarwal 19b?

**Questions:**

1. What is the state-of-the-art method for solving the considered problem? Stochastic gradient-based or? Can this work be extended to that setting? I will suggest the authors to discuss this in Conclusion.

2. Should the sentence "Note that for a general loss, the optimization (16) and (17) may diverge" below Eq (17) be revised as "Note that for a general loss, update using (17) may diverge". This sentence is unclear and less accurate because first, (16) is only an approximate loss and it's incorrect to say it may diverge.

---

> ### Author Response · Authors · 2023-11-21
> **Reply to Reviewer n99L**
>
> We thank the reviewer for the helpful feedback. We answer to individual weaknesses and questions below.
>
> - *The literature survey in this paper is not good enough. They surveyed many papers in Section 2, but the details of the most relevant papers are not provided either in the main text or in the Appendix. I'm particularly interested in existing papers that can also compute the (approximate) gradient of the function $\ell(x^\*(w))$.*
>     - In the revised version, we added a detailed description of previous related methods in Appendix B. We hope that this provides a good overview for readers who are not familiar with all the related work.
>
> - *Moreover, the authors claim that the proposed method generalizes many existing methods and discuss this in Example 1,2,3. For more general problems rather than these examples, whether the proposed method generalize existing methods?*
>     - Our method fully captures the update rules of all methods appearing in example 1-6. We do not restrict these methods to a specific case in the examples, the setups described in the examples mostly capture these methods in their full generality. Of course, beyond just the update rules used as gradient replacement, some of these methods come with deep individual theoretical results and extensions (e.g. Fenchel Young losses), which are currently not directly captured by our framework.
>
> - *From the experiment (Figure 2), even in terms of Epochs, LPGD-medium $\tau$ is faster than GD. This is not straightforward to me. If I understand correctly, all the simulated methods in the experiments are based on GD, while the gradient is computed based on different approaches. Therefore, I wonder whether the poor performance of GD comes from the low accuracy of gradient information computed by the method in Argarwal 19b?*
>     - It is important to mention that LPGD, in general, does not compute the true gradient that is used by the gradient descent baseline. We deliberately decide to use finite values of $\tau$ which intuitively means that the solver is not linearly approximated when computing the parameter updates. Not doing this approximation allows for more informative updates containing higher-order information. This is why in our experiment, the updates computed by LPGD lead to better results than plain GD. We do not believe that the baseline Agrawal19b suffers from low accuracy of gradient information, as the linear system which has to be solved during their backward pass to compute the gradient, is solved to a very high accuracy.
>     -  However, note that our method LPGD is also capable of computing the true gradient when considering the limit $\tau\rightarrow 0$, as shown in Theorem 5.1.
>
> - *What is the state-of-the-art method for solving the considered problem? Stochastic gradient-based or? Can this work be extended to that setting? I will suggest the authors to discuss this in Conclusion.*
>     - The state of the art method highly depends on the specifics. This includes: 1) Is the parameter-to-solution mapping smooth? (SOTA: stochastic gradient descent, gradients computed via implicit function theorem or direct loss minimization), otherwise 2) Are ground truth solutions to the optimization problem or objective parameters available? (SOTA: smart predict then optimize setting) 3) Is it possible to solve a regularized version of the problem? (SOTA: Fenchel Young losses) 4) Is an efficient projection method available? (SOTA: Identity with projection) 5) Is the solver only available as a blackbox oracle and no direct supervision on cost parameters/solutions? (SOTA: blackbox backpropagation)
>     - With our unification we show that for these various settings, there is actually one generalized method that performs well over many different scenarios. We think that this is an important part of the contribution, and we emphasized it more in the revised conclusion.
>
> - *Should the sentence ``Note that for a general loss, the optimization (16) and (17) may diverge" below Eq (17) be revised as "Note that for a general loss, update using (17) may diverge". This sentence is unclear and less accurate because first, (16) is only an approximate loss and it's incorrect to say it may diverge.*
>     - Thank you for pointing this out, we changed the assumption to the existence of a non-empty solution set in the revision.

---

> > ### Comment · Reviewer_n99L · 2023-11-22
> > **Thank you very much for your reply**
> >
> > Dear authors,
> >
> > I would like to thank you for your efforts in revising the paper and replying to my comments. I agree to improve the score.
> >
> > Best regards

---

### Official Review · Reviewer_m9w5 · 2023-10-29

**Soundness:** 3 good
**Presentation:** 3 good
**Contribution:** 2 fair
**Rating:** 5
**Confidence:** 3

**Summary:**

The main contribution of this paper is the unification of several prior methods into a comprehensive framework termed Lagrangian Proximal Gradient Descent (LPGD). This approach is rooted in traditional proximal optimization techniques. The study further delves into the application of LPGD in different scenarios, emphasizing its potential improvements over gradient descent in certain settings namely, those where the gradient is uninformative (e.g. discrete optimization).

**Strengths:**

Overall, the paper is well-written and easy to follow. The contribution is of interest for the community.

**Weaknesses:**

There are two main weaknesses:

1) The experiment section is rather small compared to the rest of the presentation. A lot of mathematical details could be eluded or referred to by a citation to a previous work (e.g. derivation of Moreau envelope, proximal operators, etc..). A quick (and rigorous) definition is more than enough, see below for a few remarks.

2) There is too much context at the beginning of the presentation. No need to mention a vector $\mu$ that is not used in the demonstration. I would first consider a convex optimization problem, and introduce the Lagrangian and the primal/dual variables. Explain that $f$ can be composite and have a non-smooth component, and explain Moreau and its connection with the proximal operators with 2 definitions. Finally, you can start your demonstration. Some space would be saved by shortening in "Background and Notations" section can be used to do one more experiment (see above).

=========

A few remarks:
- Missing hypotheses in the Moreau envelope definition: $f$ must be proper and lower semi-continuous.
- Instead of establishing a connection between the Moreau envelope and the proximal operator, refer the reader to a chapter in a well-known textbook that already explains in great detail the connection (e.g. Bauschke and Combettes).

=========

Overall, I believe this work deserves to be published in a venue like ICLR. However, in its current form, the manuscript is not ready for publication. The authors need to put more effort into establishing a more concise background and more rigor in the definitions of the notions used. Furthermore, the experiments section should be enriched with real-world data.

**Questions:**

1) In your experiments figure (figure 2), how do you explain that the $LPGD$ (avg) yields a significantly lower loss compared to the $LPGD$ (lower) and $LPGD$ (upper) schemes? Could you provide an intuition?

2) Could you elaborate on the convergence rate of LPGD? Is is the same as vanilla gradient descent with the additional cost of evaluating the upper and lower proximal maps?

---

> ### Author Response · Authors · 2023-11-21
> **Relpy to Reviewer m9w5 (1/2)**
>
> We thank the reviewer for the helpful suggestions. We answer to the individual weaknesses and questions below.
>
> - *The experiment section is rather small compared to the rest of the presentation.*
>     - The main contribution of this paper is in showing the similarities between different methods developed for different experimental settings, and how they can be combined in a unified framework. We believe that this contribution is decoupled from experimental results, and does not need to be supported by experiments, as each of the methods has previously been established independently to achieve strong empirical performance.
>     - However, in our mission of unification we noticed that our framework actually suggests applying finite-difference based methods (with finite values of $\tau$) even in the setting in which the solver has non-degenerate derivatives. We tested this on our small-scale toy-experiment and observed that this technique can indeed improve performance over the previously typically used gradient-based approaches. Exploring this in-depth is beyond the scope of our paper, but we still wanted to include the small experimental section to share that our method could also lead to empirical improvements in future work.
>
> - *A lot of mathematical details could be eluded or referred to by a citation to a previous work (e.g. derivation of Moreau envelope, proximal operators, etc..). A quick (and rigorous) definition is more than enough, see below for a few remarks.*
>     - We changed many parts of the paper in the revision to make the treatment more rigorous by e.g. clearly stating the assumptions.
>
> - *There is too much context at the beginning of the presentation. No need to mention a vector $\mu$ that is not used in the demonstration. I would first consider a convex optimization problem, and introduce the Lagrangian and the primal/dual variables. Explain that $f$ can be composite and have a non-smooth component, and explain Moreau and its connection with the proximal operators with 2 definitions. Finally, you can start your demonstration. Some space would be saved by shortening in "Background and Notations" section can be used to do one more experiment (see above).*
>     - We rewrote entirely the problem setup section, which now presents more clearly what our aim is and which components are involved in the learning algorithm. Unfortunately the vector $\mu$ is needed here when trying to write down the training objective (equation (3) in revision).
>     - Regarding the background on Moreau envelope and proximal map we removed the unnecessary part on the intuition and instead added references as suggested by the reviewers, we also removed the relation between proximal gradients and projected gradients. However, we still think that it is important to give some background on the proximal point method and proximal gradient descent here as these algorithms are so closely connected to our Lagrangian versions of them.
>
> - *Missing hypotheses in the Moreau envelope definition: $f$ must be proper and lower semi-continuous.*
>     - Thank you for pointing this out, we added this in the revision.
>
> - *Instead of establishing a connection between the Moreau envelope and the proximal operator, refer the reader to a chapter in a well-known textbook that already explains in great detail the connection (e.g. Bauschke and Combettes).*
>     - We included the additional reference and restructured the section as described above.
>
> - *In your experiments figure (figure 2), how do you explain that the $LPGD$ (avg) yields a significantly lower loss compared to the $LPGD$ (lower) and $LPGD$ (upper) schemes? Could you provide an intuition?*
>     - This is intuitively similar to the central-difference method having higher precision than forward or backward difference methods. Further intuition can be gained from Figure 1, which highlights that the different envelopes provide non-zero gradients of the envelope in different subsets of the domain, while $LPGD$ (avg) is able to use informative gradient information in both subsets. We added this intuition to the results section of the revision.

---

> > ### Author Response · Authors · 2023-11-21
> > **Reply to Reviewer m9w5 (2/2)**
> >
> > - *Could you elaborate on the convergence rate of LPGD? Is is the same as vanilla gradient descent with the additional cost of evaluating the upper and lower proximal maps?*
> >     - The convergence rate of LPGD and GD is not the same, as LPGD does not compute the original gradients.
> >     - While we have some additional overhead computation required to evaluate the upper and lower Lagrangian proximal map (for which in LPGD we can use the forward solver oracle, which can be much faster on the backward pass due to warm-starting), the computed updates carry more information than gradients. Intuitively, gradients result from a first order linearization of the solution mapping, our method is able to compute higher order information updates by instead running the solver oracle again. This can lead to much faster convergence as seen in the experiment.
> >     - In this work we did not attempt to derive a convergence rate, which would require stronger assumptions (e.g. currently we do not assume convexity of the loss or the backbone). However, it will be an interesting direction for future research, as the results would immediately translate to the various methods that we generalize.
> >
> > We hope that our reply answers the reviewers questions and that the various updates to the paper during the revision resolve some of the weaknesses mentioned by the reviewer.

---

> > > ### Comment · Reviewer_m9w5 · 2023-11-22
> > >
> > > I thank the authors for their comprehensive answers to my concerns. I raised my score.

---

### Official Review · Reviewer_R9P5 · 2023-11-01

**Soundness:** 2 fair
**Presentation:** 1 poor
**Contribution:** 2 fair
**Rating:** 3
**Confidence:** 4

**Summary:**

This paper focuses on a learning model with parameter $\theta$ that produces another set of parameters $w$ that defines a primal-dual optimization problem with solution $z^* = (x^*, y^*)$. The goal is to optimize the whole pipeline with respect to the problem defining parameters $w$ that can be then connected to model parameter $\theta$. Since with respect to $w$, this is a non-smooth problem, the authors design a framework inspired by classical proximal gradient method (PGM). The new framework is named Lagrangian PGM since a "divergence" based on difference of Lagrangians is used to define the gradient-descent type update. The authors show an asymptotic result that as the coefficient in front of the Lagrangian "divergence" term approaches to 0, the true gradient is achieved.

**Strengths:**

The problem considered in the paper and the motivation are interesting (albeit difficult to understand in the current presentation). The authors are making connections with classical approaches in nonsmooth optimization such as Lagrangian, Moreau envelopes, mirror descent and Bregman divergences to get inspiration to tackle the specific nonsmooth optimization problem at hand. The subtle and important aspects such as assuming the existence of unique solutions are pointed out and some effort has been made to partially address these shortcomings. It is also good to see the authors discussing the limitations in Appendix A with preliminary ideas on how one might go around improving on them.

**Weaknesses:**

Unfortunately, the paper is not written in a precise way for problem statements, results, tools etc. In many places it is not clear what the goal is and hence the contribution of the paper is quite unclear both in terms of mathematics and also practical aspects (not clear how to solve subproblems etc.). The assumptions and discussions about them are unclear. Even though many technical concepts are introduced, not much is proven with them and much of the claims in the paper are not proven (a list is below). The problem definition, goal etc are not clearly written and it requires the reader to read multiple times to even understand this. Unfortunately, the presentation needs to be vastly improved for this paper to be suitable for readers to understand and gain from.

- Proper references are not given in many places. For example, last paragraph in page 1 cites for "traditional proximal optimization" the papers by Moreau, 1962 (an excellent reference) and Parikh&Boyd, 2014, Boyd&Vandenberghe, 2004 which are okay (even though it is not clear where in Boyd&Vandenberghe's book proximal methods are discussed) however many of the most important references are not there. For example, no mention of Rockafellar who is the founding father of the field along with Moreau, his book Convex Analysis (at the least) should be cited here with many of his other founding papers. One of the most impactful proximal gradient based algorithms is FISTA by Beck and Teboulle which is also not cited. Nesterov's or Tseng's accelerated proximal methods are not cited. Another important reference is by Figueiredo, Nowak, Wright, 2007 which is again, not cited. It is also not hard to find these references, they are all included in Parikh & Boyd's text.

- Sect 3.1, problem setup is not written well. One can write this in a much clearer fashion. Especially for optimization audiences, it is much better to write a well-defined problem, rather than the arrows and unclear notation in the start of Sect 3.1. For example, one should define precisely the "model" $W_\theta$. What kind of a function is it? What assumptions are needed on it? One then needs to write the optimization problem in terms of the main variable ($w$ or $\theta$) and then clearly describe the assumptions on each function appearing in the problem.

- Problem (1) is well-defined but it depends on $w$ hence it does not really describe the whole problem. It should be written in a precise problem formulation by taking into account the relations with arrows given at the start of the section.

- For problem (1) the authors assume that unique solution exists, which is, of course a very strong assumption. The authors say they address this point in App F, which in my reading goes to a tangent about "graphical derivatives" rather than showing us how to get around this assumption. In particular, how to generalize the results in the paper without assuming unique solution? A precise statement is needed rather than an extended discussion which is both difficult to follow and also difficult to follow to other parts of the paper. What would be good is to point out how the particular statements in the paper will change when one considers non-unique solutions etc. Right now, this is not very understandable unfortunately.

- Sect 4.2 requires eq. (14, 15) are uniquely attained, why would this be true? App F or G does not really show this (or even if it does, it is very unclear). Please provide precise statements and then proofs in App F and G. Right now it is unclear what the authors are trying to show.

- It is not clear how one actually implements the algorithms described in the paper. In fact, the algorithms are not even written down, only gradients are derived for example in eq. (28, 29).  I understand the algorithm is just gradient descent but the gradients depend on many other things such as $z^*$ and $\tilde z^\tau$ that the reader needs to track down in the quite heavy text. They need to be presented in a compact fashion to be able to see what the algorithm is. Next, lookng at eq. (28, 29), how does one solve for $z^*$ and $\tilde z^\tau$? The authors mention here "same efficient optimization algorithm used to solve (1)": what is this algorithm? What kind of an algortihm are we lookng for? What is its cost? Clearly, one cannot solve the problem to exact solution, how does one tolerate inexactness?

- Example 2 again refers to Appendix F for an "in-depth" discussion whereas the discussion in Appendix F is quite hard to connect to things in the main text. As I asked above, App F and G should be more precise with statements and proofs and connections to the main text.

- Where in App G the proof for Theorem 4.1 with non-unique solutions given?



+++ post-rebuttal

I already explained the reasons for keeping my score unchanged in my follow-up message. Here are some more remarks for the authors so they can hopefully improve their manuscript in the future. --  the main result as stated by the authors is essentially stating that asymptotically they can approximate the gradient. This result is not a satisfactory convergence result in my reading. It is unclear to me how the authors would extract meaningful results from here. I do not believe that results in this form where the implications or meaning are not clearly explained help the community, but I think such practice cause only confusion. Please give an effort to explain your results to show what they can and **more importantly** what they cannot do. -- I am not against the improvement of the paper during review stage and totally acknowledge this is one of the most important aspects of the review process. However, when the arguments of the paper needs to be significantly modified as is the case here and these modifications are not clear to the reviewers (not clear to me in this case), I do not think that we can do a healthy evaluation in such a short time frame. There is a limit to the amount of changes that can be made to a paper after which the paper needs a new conference review cycle. I am not sure that it is fair to the reviewers that the authors submit a version of their manuscript with incorrect or imprecise claims (as pointed out in my review and accepted by the authors) and then expect the reviewers to do another review to the paper with major modifications in a short discussion period (until Dec 5th). In summary, I am not claiming the analysis is not correct (the next review cycle should help clarify this), but I am saying that even if it is correct, the results do not seems satisfactory to me and the paper, in my view, is not written in an accessible fashion. Please refer to my review for examples.

**Questions:**

More questions also appear in "Weaknesses" section of my review.

- Last paragraph of page 1 states that the paper can solve problems with non-linear objectives or learnable constraints. First, it is not clear what "learnable constraints" mean. Second, the paper itself says in Sec 4.3 that eq. (16) and (17) might "diverge" and hence they focus on linear loss in Sec 4.5. How does the paper handles non-linear loss functions? Also, what does it mean for (16) a value of an optimization problem and (17) the optimizer of the problem to "diverge"? Do the authors mean that a solution might not exist? Or do the authors mean that an algorithm might diverge for solving the problem? It is quite unclear.

- What is the purpose of introducing $D_2$ in eq. (5)? clearly it is just the Euclidean squared norm. I can see that the authors generalize later to a "divergence" in Sect 4.1 but $D_2$ notation is not really necessary here given that the paper has so many other notations making it difficult to follow for a reader.

- With Moreau smoothing, we have an extended theory on the smoothing affect, but the authors just replace the squared norm with the "Lagrangian divergence" and state that now we have a smoothed objective. This needs to be proven. Why is this envelope smooth? With which Lipschitz constant? What does one need to assume about the Lagrangian to be able to argue smoothness of the envelope? For sure justification is needed since as the authors also cite in App C, Bauschke et al., 2018 did a systematic study with Bregman Moreau envelopes. Such a precise statement justifying "smoothness" is missing.

- eq. (12): bad notation, x on the left and right hand sides are not the same.

- eq. (13): either needs to be proven or given a reference.

- Where are the proofs of Prop 4.2 and 4.3?

- Sect. 4.4: why are the assumptions of Danskin's theorem satisfied in this case, can the authors please clarify?

- Sect. 4.5: what does it mean to "expose the linear parameters of the Lagrangian" right before eeq. (25)? Where is $\Omega$ defined?

---

> ### Author Response · Authors · 2023-11-21
> **Reply to Reviewer R9P5 (1/3)**
>
> We thank the reviewer for the constructive criticism. We agree with many of the weaknesses regarding the original manuscript, and significantly improved the presentation and quality in the revised version. This includes, among many other changes, a clarified problem formulation with updated and precisely formulated assumptions, proofs and corrections for the claims in the paper, and a clarified presentation of the implementation of our method including an algorithm box. We provide more detailed answers to the individual points of critique below.
>
> - *Unfortunately, the paper is not written in a precise way for problem statements, results, tools etc. In many places it is not clear what the goal is and hence the contribution of the paper is quite unclear both in terms of mathematics and also practical aspects (not clear how to solve subproblems etc.). The assumptions and discussions about them are unclear. Even though many technical concepts are introduced, not much is proven with them and much of the claims in the paper are not proven (a list is below). The problem definition, goal etc are not clearly written and it requires the reader to read multiple times to even understand this. Unfortunately, the presentation needs to be vastly improved for this paper to be suitable for readers to understand and gain from.*
>     - In the revision we make the assumptions more precise (both in the problem statement and in the proofs). We also include additional proofs in a more standard format in the revised Appendix F.
>     - Regarding the unclear main contribution of our work, we want to stress that it is in highlighting similarities between various methods for embedding paramterized optimization problems in machine learning architectures and showcasing that the corresponding update rules can be derived from principles routed in traditional optimization algorithms. We made this goal clearer in the revision, see e.g. the changes in the abstract and introduction.
>
> - *Proper references are not given in many places. For example, last paragraph in page 1 cites for "traditional proximal optimization" the papers by Moreau, 1962 (an excellent reference) and Parikh\&Boyd, 2014, Boyd\&Vandenberghe, 2004 which are okay (even though it is not clear where in Boyd\&Vandenberghe's book proximal methods are discussed) however many of the most important references are not there. For example, no mention of Rockafellar who is the founding father of the field along with Moreau, his book Convex Analysis (at the least) should be cited here with many of his other founding papers. One of the most impactful proximal gradient based algorithms is FISTA by Beck and Teboulle which is also not cited. Nesterov's or Tseng's accelerated proximal methods are not cited. Another important reference is by Figueiredo, Nowak, Wright, 2007 which is again, not cited. It is also not hard to find these references, they are all included in Parikh\&Boyd's text.*
>     - Thank you for pointing out these missing references; we include them in the revision.
>
> - *Sect 3.1, problem setup is not written well. One can write this in a much clearer fashion. Especially for optimization audiences, it is much better to write a well-defined problem, rather than the arrows and unclear notation in the start of Sect 3.1. For example, one should define precisely the "model" $W_\theta$. What kind of a function is it? What assumptions are needed on it? One then needs to write the optimization problem in terms of the main variable ($w$ or $\theta$) and then clearly describe the assumptions on each function appearing in the problem. Problem (1) is well-defined but it depends on $w$ hence it does not really describe the whole problem. It should be written in a precise problem formulation by taking into account the relations with arrows given at the start of the section.*
>     - We completely restructured Section 3.1. It now includes a well-defined problem formulation and lists the assumptions that we make.

---

> > ### Author Response · Authors · 2023-11-21
> > **Reply to Reviewer R9P5 (2/3)**
> >
> > - *For problem (1) the authors assume that unique solution exists, which is, of course a very strong assumption. The authors say they address this point in App F, which in my reading goes to a tangent about "graphical derivatives" rather than showing us how to get around this assumption. In particular, how to generalize the results in the paper without assuming unique solution? A precise statement is needed rather than an extended discussion which is both difficult to follow and also difficult to follow to other parts of the paper. What would be good is to point out how the particular statements in the paper will change when one considers non-unique solutions etc. Right now, this is not very understandable unfortunately.*
> >     - We agree that this was not written in an very understandable way. In the revised version we change our assumption of the uniqueness of the optimizer to assuming the existence of a selection of the solution set continnuous at $w$, which holds whenever the true derivative exists. With this assumption we do not require the separate treatment of unique vs non-unique as before, and therefore the corresponding appendices are removed in the revision.
> >
> > - *Sect 4.2 requires eq. (14, 15) are uniquely attained, why would this be true? App F or G does not really show this (or even if it does, it is very unclear). Please provide precise statements and then proofs in App F and G. Right now it is unclear what the authors are trying to show.*
> >     - The old manuscript just assumed the uniqueness of the optimizer, it was not attempting to prove this. In the revision we instead have to assume the existence of a continuous selection from the solution set of eq. (13,14,15,16).
> >
> > - *It is not clear how one actually implements the algorithms described in the paper. In fact, the algorithms are not even written down, only gradients are derived for example in eq. (28, 29). I understand the algorithm is just gradient descent but the gradients depend on many other things such as $z^\*$ and $\tilde{z}^\tau$ that the reader needs to track down in the quite heavy text. They need to be presented in a compact fashion to be able to see what the algorithm is. Next, lookng at eq. (28, 29), how does one solve for $z^\*$ and $\tilde{z}^\tau$? The authors mention here "same efficient optimization algorithm used to solve (1)": what is this algorithm? What kind of an algortihm are we lookng for? What is its cost? Clearly, one cannot solve the problem to exact solution, how does one tolerate inexactness?*
> >     - We add an algorithm box in the revision to show how we replace the co-derivative computation during gradient backpropagation. We also clarify how this algorithm ties into the automatic differentiation framework by replacing the backward pass (end of Section 3 and 5.4).
> >     - Regarding the question on how to solve for $z^\*$ and $\tilde{z}^\tau$, we are concerned with the case in which a user has an existing efficient algorithm for solving a parameterized problem and wants to use e.g. a neural network to predict the parameters of the optimization problem that this solver optimizes. We are not concerned with how exactly this solver should look like, in our experiments it is a conic program solver. Again, we agree that this reasoning was not clearly presented in the original text and we made an effort to clarify it in the revision, see e.g. Section 5.4.
> >     - The question of how much we can tolerate inexactness of the solver is interesting. We suspect that for small values of the hyperparameter $\tau$ the inexactness can become problematic as the perturbed solution approaches the original solution. However, we anyway advocate for large $\tau$ instead, and in our experiment we are also able to solve the optimization problems to a high accuracy and therefore did not observe issues arising from inexactness. We clarify the paragraph on this potential limitation in Appendix A of the revision.
> >
> > - *Example 2 again refers to Appendix F for an "in-depth" discussion whereas the discussion in Appendix F is quite hard to connect to things in the main text. As I asked above, App F and G should be more precise with statements and proofs and connections to the main text.*
> >     - These appendices are removed in the revision.
> >
> > - *Where in App G the proof for Theorem 4.1 with non-unique solutions given?*
> >     - We remove the distiction between unique vs non-unique in the revised version. The proof of the Theorem in its new generality is provided in Appendix F of the revision. This appendix now also contains the proofs for the remaining statements.

---

> > > ### Author Response · Authors · 2023-11-21
> > > **Reply to Reviewer R9P5 (3/3)**
> > >
> > > - *Last paragraph of page 1 states that the paper can solve problems with non-linear objectives or learnable constraints. First, it is not clear what "learnable constraints" mean.*
> > >     - By learnable constraints we mean that our problem formulation covers saddle-point problems that can be used to represent parameterized constraints on the primal variables. This means that we can compute informative updates for the parameters of these constraints, therefore making them learnable with gradient descent. We updated the text to make this more precise.
> > >
> > > - *Second, the paper itself says in Sec 4.3 that eq. (16) and (17) might "diverge" and hence they focus on linear loss in Sec 4.5. How does the paper handles non-linear loss functions?*
> > >     - We mostly handle non-linear loss functions by linearizing them around the optimizer of the forward problem (1), the approach we refer to as LPGD. This allows us to re-phrase the backward problem (Lagrangian proximal map (13,15)) in a form that can be solved by the solver oracle.
> > >     - However, we also note that in principle the backward problem (13,15) could be solved by optimization algorithms other than the forward solver oracle. But in this paper the main focus is on the LPGD algorithm, therefore we do not explore this direction in more detail.
> > >     - In the revision we clarify this distinction between LPPM and LPGD in the beginning of Section 5.4.
> > >
> > > - *Also, what does it mean for (16) a value of an optimization problem and (17) the optimizer of the problem to "diverge"? Do the authors mean that a solution might not exist? Or do the authors mean that an algorithm might diverge for solving the problem? It is quite unclear.*
> > >     - Indeed we wanted to say that a solution might not exist, the use of the term "diverge" was not correct here, thanks for pointing this out. We now assume the non-emptyness of the solution set instead. A sufficient condition is e.g. the compactness of $\mathcal{X}$ and $\mathcal{Y}$ which is often the case (e.g. in all of our examples considering an LP).
> > >
> > > - *What is the purpose of introducing $D_2$ in eq. (5)? clearly it is just the Euclidean squared norm. I can see that the authors generalize later to a "divergence" in Sect 4.1 but $D_2$ notation is not really necessary here given that the paper has so many other notations making it difficult to follow for a reader.*
> > >     - We removed this extra definition in the revision. We understand that the notation is quite heavy, but we made an effort to keep it as intuitive as possible and included a list of symbols with links to the definitions in Appendix H to help the readers.
> > >
> > > - *With Moreau smoothing, we have an extended theory on the smoothing affect, but the authors just replace the squared norm with the "Lagrangian divergence" and state that now we have a smoothed objective. This needs to be proven. Why is this envelope smooth? With which Lipschitz constant? What does one need to assume about the Lagrangian to be able to argue smoothness of the envelope? For sure justification is needed since as the authors also cite in App C, Bauschke et al., 2018 did a systematic study with Bregman Moreau envelopes. Such a precise statement justifying "smoothness" is missing.*
> > >     - Unfortunately we were not able to proof such a statement, we only have visualizations as in Figure 1 to support our intuition here. We therefore remove the claim that the envelopes are smooth in the revision.
> > >
> > > - *eq. (12): bad notation, x on the left and right hand sides are not the same.*
> > >     - Fixed in the revision.
> > >
> > > - *eq. (13): either needs to be proven or given a reference. Where are the proofs of Prop 4.2 and 4.3?*
> > >     - We now provide these proofs in Appendix F. We also make the assumptions more precise.
> > >
> > > - *Sect. 4.4: why are the assumptions of Danskin's theorem satisfied in this case, can the authors please clarify?*
> > >     - Thank you for pointing this out, the assumptions for Danskin's theorem were indeed not satisfied. We now rely on a different result by Oyama & Takenawa (2018, Proposition 4.1). We also restate many of our assumptions based on this transition. For example, we now assume the existence of a continuous selection of the solution mapping, but instead require less assumptions on both the Lagrangian and the optimization problem (1).
> > >
> > > - *Sect. 4.5: what does it mean to "expose the linear parameters of the Lagrangian" right before eeq. (25)? Where is $\Omega$ defined?*
> > >     - This was again not presented clearly enough. We actually assume a specific form of the Lagrangian here in which it has a parameterized term that is linear in $x$. We make this more rigorous in the revision.
> > >
> > > We again thank the reviewer for the very helpful critical feedback, which allowed us to improve the paper a lot during the revision.

---

> > > > ### Comment · Reviewer_R9P5 · 2023-11-22
> > > > **Follow-up**
> > > >
> > > > I thank the authors for their revision and the attempt to address my comments. Even though things get clearer, unfortunately there are still so many aspects of the paper that needs to be improved.
> > > >
> > > > - The extensive changes will need to be checked thoroughly in a proper review cycle, instead of a short post-rebuttal period (from the time of the rebuttal to the end of discussions was 2 days for this submission). This is because fundamental aspects of the paper has changed. The authors move from unique/non-unique solution treatment to using continuous selection from solution set. Now they no longer argue smoothness (in view of my comments) where the first version was all about "smoothed envelopes", without smoothing, solving subproblems become even harder of course. Danskin's theorem's central use was not correct and now they cite a paper by Oyama, Takenawa, 2018 that I am not aware and the new argument should be thoroughly checked.
> > > >
> > > > - The results (at least what they mean) are still unclear. Theorem 5.1 is okay, we are asymptotically estimating the gradient. However, Proposition 5.2 and 5.3 are still unclear. We are interested in an end-to-end pipeline but the result is for fixed $w$. We know that for a fixed $w$, the prob (1) is solvable, this is the primitive that the paper relies on. But then what do these propositions mean for a fixed $w$? Fixed $w$, we already have guarantees. The guarantees should be clearly explained (what we get and we cannot get from them) in line with the motivation of the paper (we are not interested in a solution for a fixed $w$, but we are interested in the whole learning pipeline). For a theoretical treatment, more precision is needed. It is not clear how significant the results are because of this.
> > > >
> > > > - The algorithm in Alg 1 is still unclear. I asked about how to estimate $z^*$ and other values, algorithm just says "load $z^*$" with no reference to where it is even defined. With so many definitions, it is so hard to follow these things to see what the primitives of the algorithm are (what kinds of solvers one would need and what the solvers would provide), what the precise results are etc.
> > > >
> > > > - Role of inexactness for subproblems is not investigated. The authors say in their rebuttal it is not clear how to handle this in the theory. But of course we are not going to get exact solutions by solving a min-max problem (the solvers are even slower without smoothness), especially with large-scale problems. This is an important point that needs to be addressed in such a paper. We are lacking an end-to-end algorithmic pipeline here.
> > > >
> > > > In summary, unfortunately, I still do not think that the results are clear or strong enough yet for publication. Of course it is a positive that the authors incorporated feedback and fixed incorrect points in their paper, but this brings into question the state of the paper when it was submitted. With the significant amount of changes made in the revision stage, the paper needs to go through a proper review cycle rather than 2 days for verifying the new claims.  I am sorry but these are the reasons that I have to keep my score as is.

---

> > > > > ### Author Response · Authors · 2023-11-22
> > > > > **Reply to Follow-up**
> > > > >
> > > > > We thank reviewer R9P5 for answering our rebuttal and positively acknowledging that we incorporated feedback and fixed incorrect points in the revision.
> > > > >
> > > > > We agree that the substantial changes we made in the revision also require being thoroughly checked. Even though the discussion period with author involvement ends today, the reviewer-AC discussion still continues until December 5.
> > > > > Therefore, if there are some remaining issues with our use of the result by Oyama & Takenawa (2018), reviewer R9P5 should have opportunity to raise them. Otherwise, the corresponding concern in the original review appears to be adequately resolved.
> > > > >
> > > > > The reviewer agrees that our central Theorem 5.1 is now ok. Propositions 5.2 & 5.3 are not central to the functionality of our theory; they only investigate a particular choice of parameter $\tau$ for one subproblem, i.e. solving the lower Lagrangian proximal map. The propositions show that for this particular choice, the map reduces to known problems such as Frank-Wolfe iterations or standard proximal gradients. Elucidating these links (which we then use in the following examples) is the only purpose of the propositions.
> > > > >
> > > > > Concerning reviewer R9P5's comments about algorithm 1, we are puzzled:
> > > > > 1) BackwardPass reads "load $z^*$ from forward pass"
> > > > > 2) ForwardPass contains "$z^* \leftarrow \text{SolverOracle}(w)$, Solve optimization $(2)$" with $(2)$ being the equation defining $z^*(w)$
> > > > > 3) After equation (2) in Problem Setup we write: "Throughout the paper, we assume access to an oracle that efficiently solves (2) to high accuracy. In our experiments, (2) is a conic program which we solve using the SCS solver (O’Donoghue et al., 2016)."
> > > > >
> > > > > We believe it should be clear where $z^*$ in BackwardPass originates.
> > > > >
> > > > > Regarding the role of the inexactness of solutions to subproblems, we are trying to present our framework in as much generality as possible. The most general assumption that allows this is to assume a solver that returns a high-accuracy solution (continuous at $w$). Clearly, this is not possible for all optimization problems that could be stated in the form of equation (1). However, presented in this form, we capture e.g. large-scale convex-concave problems or, alternatively, even small-scale but highly non-convex problems. Characterizing the vast amount of optimization problems of the form (1) as those problems that satisfy our assumptions and those that do not is very difficult and beyond the scope of this work; therefore, we resort to examples instead.
> > > > >
> > > > > Finally, we would like to again thank the reviewer for the time invested in reviewing our work. We believe the paper strongly benefits from the detailed initial comments.

---

### Author Response · Authors · 2023-11-20
**Answer to all Reviewers**

We thank the reviewers for the constructive criticism and helpful suggestions. We summarize the main points of critique below along with the corresponding changes we made in the revision.

- Clarity and precision of paper: (reviewer R9P5)
    - In our revision we rewrote significant parts of the paper to make our presentation more precise.
    - We rewrote the problem setup (Section 3) as well as the abstract to make the problem definition/goal/contribution clearer. We also added an algorithm box to compactly show how LPGD works.
    - We modified our assumptions and clearly state them (Section 3, 5.1, 5.4). A major change in the assumptions is that we assume the existence of a contiuous selection of the solution mapping instead of assuming a unique optimizer. We also removed the assumptions on convexity which are not required in our proofs, and therefore we renamed the title to reflect this.
    - We added Appendix F which clarifies the previous proof of the Theorem and contains new proofs for the other technical claims in the paper.
- The experiment section is too small, presentation of problem setup & background could be shortened (reviewer m9w5)
    - We shortened the background section and rewrote the problem setup in a more concise and understandable way. However, we do not add additional experiments as the main contribution of this work is in the derivation of a unifying framework for existing methods, not in the application of this framework to new experimental setups.
- It is unclear why LPGD achieves better performance than GD (reviewers m9w5 and n99L)
    - We clarify in the revision that LPGD does not compute the same updates as GD, and better present how exactly LPGD differs from standard gradient descent (Section 5.4, Algorithm 1). We also extended the intuition of why LPGD achieves better performance than GD in practice (Section 6).
- The literature survey is not good enough, the most related methods are not explained in enough detail (reviewer n99L)
    - We added a more detailed description of the most related methods in Appendix B.
- We do not provide a convergence rate analysis (reviewers m9w5 and t8qX)
    - This unfortunately has not changed in the revision. This paper tries to make as few assumptions as possibly needed to justify the use of our framework for a large class of problems (e.g. no assumptions on convexity of loss, Lagrangian, or backbone model). However, we believe that under stronger assumptions our unifying framework can potentially open the door for a unified analysis of convergence for the variety of methods captured by our framework.

We hope that the significant changes we made to the manuscript help resolve the criticism on the original draft. We answer to the individual questions and weaknesses below.

---

### Meta-Review · Area_Chair_trBU · 2023-12-03

**Metareview:**

This paper studies a Lagrangian proximal gradient method for learning convex optimization models. As pointed by some reviewers, the paper may not be ready to be published in its current form. The original submission has many issues in terms of clarity and significance of the results. During the rebuttal period, the authors uploaded a major revision, which is almost a completely rewritten paper. This brings a lot of concerns regarding the correctness and presentation of the original submission. Such a major revision will need to go through a new review cycle, which is not possible for the reviewers in a short time period. Moreover, many issues in the paper still persist in the revision. We therefore cannot accept this paper in its current form. The authors need to submit to a later venue to go through another review process.

**Justification For Why Not Higher Score:**

The paper has many issues. The revision almost completely rewrote the entire paper. Such major changes cannot be reviewed thoroughly by the reviewers in a short time period.

**Justification For Why Not Lower Score:**

NA

---

### Decision · Program_Chairs · 2024-01-16

Reject